# DreamDistribution: Learning Prompt Distribution for Diverse In-distribution Generation

**Brian Nlong Zhao**[1]\*, **Yuhang Xiao**[1]\*, **Jiashu Xu**[2]\*, **Xinyang Jiang**[3], **Yifan Yang**[3],
**Dongsheng Li**[3], **Laurent Itti**[1], **Vibhav Vineet**[4]†, **Yunhao Ge**[1]†

[1]University of Southern California  [2]Harvard University
[3]Microsoft Research Asia  [4]Microsoft Research Redmond
\**Equal Contribution*  †*Equal Advising*

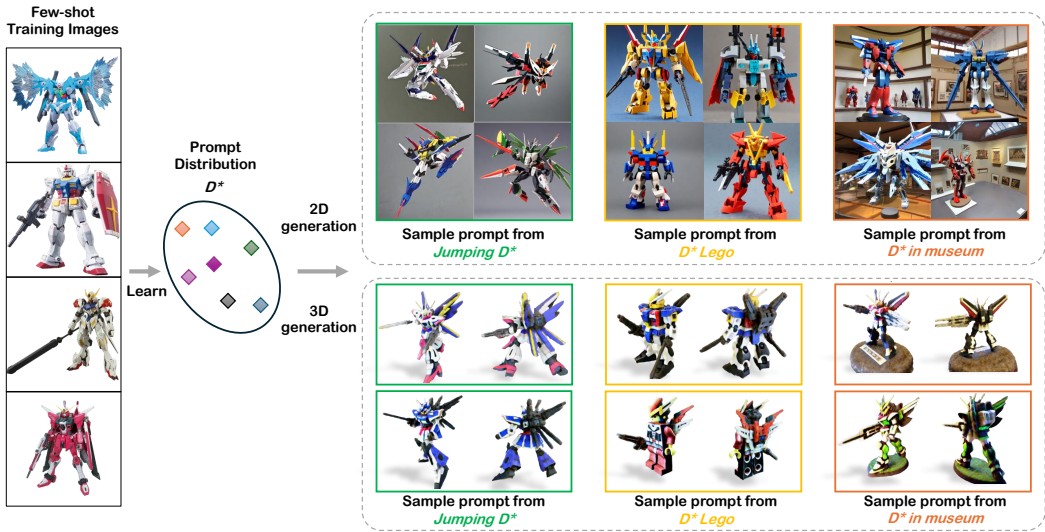

Figure 1: DreamDistribution learns a prompt distribution $D^*$ that represents a distribution of descriptions corresponding to a set of reference images. We can sample new prompts from $D^*$ or modified $D^*$ by text-guided editing to generate images of diverse new instance that follows the visual attributes of reference training images (top). We can also apply a learned distribution flexibly to, for example, a pretrained text-to-3D model, and generate diverse new 3D assets following the reference images (bottom).

## Abstract

The popularization of Text-to-Image (T2I) diffusion models enables the generation of high-quality images from text descriptions. However, generating diverse customized images with reference visual attributes remains challenging. This work focuses on personalizing T2I diffusion models at a more abstract concept or category level, adapting commonalities from a set of reference images while creating new instances with sufficient variations. We introduce a solution that allows a pretrained T2I diffusion model to learn a set of soft prompts, enabling the generation of novel images by sampling prompts from the learned distribution. These prompts offer text-guided editing capabilities and additional flexibility in controlling variation and mixing between multiple distributions. We also show the adaptability of the learned prompt distribution to other tasks, such as text-to-3D. Finally we demonstrate effectiveness of our approach through quantitative analysis including automatic evaluation and human assessment.

# 1 INTRODUCTION

Dreams have long been a source of inspiration and novel insights for many individuals (Edwards et al., 2013; Von Grunebaum & Caillois, 2023). These mysterious subconscious experiences often reflect our daily work and life (Freud, 1921). However, these reflections are not mere replicas; they often recombine elements of our reality in innovative ways, leading to fresh perspectives and ideas. We aim to emulate this fascinating mechanism in the realm of text-to-image generation.

Text-to-image (T2I) generation has recently been popularized due to the astonishing performance of state-of-the-art diffusion models such as Stable Diffusion (Rombach et al., 2021) and DALL·E 2 (Ramesh et al., 2022). Variations of the T2I models have enabled several fascinating applications that allow user to control the generation, such as conditioned generation based on other input modalities (Zhang et al., 2023; Li et al., 2023; Yang et al., 2023), inpainting (Lugmayr et al., 2022; Xie et al., 2023), image editing (Mokady et al., 2023; Brooks et al., 2023). One such interesting application is personalization of T2I models, where user provides some reference images of the same instance (*e.g.* their pet dog), and the personalized model can generate images based on the references, with the flexibility of text-guided editing for new context. This is generally achieved by associating a token with the personalized concept through fine-tuning the model parameters (Ruiz et al., 2022; Kumari et al., 2023) or newly added learnable token embeddings (Gal et al., 2022; Voynov et al., 2023).

In many cases, however, user may want to personalize T2I generation over a more abstract visual attribute instead of a specific instance-level personalization. For example, a designer may seek inspiration by generating a variety of novel cartoon characters or scenery images following similar visual attributes presented in their previous works. In this case, trying over text prompts is not scalable and hard to get desired result that follows the desired visual attributes. On the other hand, using the existing personalization methods aforementioned is likely to fail when training images do not represent the same instance, but rather encompass a distribution sharing certain, yet challenging-to-articulate, commonalities. Additionally, existing personalization methods often result in limited diversity and variation during generation (Fig. 3). Since the associated token is fixed, these methods will typically learn a token that is either overfitted to a combination of visual features, or learn a token that is overly generalized, which introduces more randomness into the uncontrollable diffusion process, thereby failing to follow desired visual attributes in generated images.

In this work, we propose DreamDistribution, a prompt distribution learning approach on T2I diffusion model for various downstream tasks (Fig. 1). Our proposed solution has three key components (Fig. 2). First, to adapt a pretrained fixed T2I model, instead of fine-tuning diffusion model parameters, our method builds on prompt tuning (Zhou et al., 2022a;b), where we use soft learnable prompt embeddings with the flexibility to concatenate with text, to associate with the training image set. This design have several advantages: (1) It prevents catastrophic forgetting of the pretrained model, enabling it to learn an almost infinite variety of target prompt distributions using the same T2I diffusion model. (2) It is highly efficient in terms of parameters, requiring only the prompt itself as the learnable element. (3) The learned prompts remain within the semantic space of natural language, offering text-guided editing capabilities and generalizing to other pre-trained diffusion models, such as text-to-3D. (4) The learned distribution increased flexibility in managing variations. Second, we introduce a distribution of prompts to model various attributes described by reference images at a broader level. The prompt distribution is modeled by a set of learnable prompt embeddings to associate with the training image set as a whole. The learned prompt distribution can be treated as a distribution of learned "descriptions" of the reference images and should be able to model the commonalities and variations of visual attributes, *e.g.*, foreground, style, background, texture, pose. During inference, we sample from the prompt distribution, which should have a similar semantic meaning, understood by the downstream denoising network, to produce in-distribution outputs with appropriate variations. Lastly, to effectively optimize the set of soft prompts that models the distribution, we apply a simple reparameterization trick (Kingma & Welling, 2013) and an orthogonal loss to update the prompts at token embedding space simultaneously and orthogonally.

We first demonstrate the effectiveness of our approach in customizing image generation tasks (§4). By taking a small set of images of interest as training images, we demonstrate that our approach can generate diverse in-distribution images where baseline methods fail to generate desired output. The diversity and the quality of our synthetic images are verified via automatic and human evaluation (§4.2). We show that the learned distribution holds the capability of text-guided editing, as well as

further controllability such as scaling the variance and composition of distributions (§4.3). Next we highlight that the learned prompt distribution can be easily applied to other text-guided generation tasks such as pretrained text-to-3D models (§4.4). Lastly we show the effectiveness of our method on personalized distribution generation through classification task with synthetic training data as a proxy (§4.5). In summary, our contributions are:

- We propose a distribution-based prompt tuning method for diverse personalized generation by learning prompt distribution using T2I diffusion model.
- Using a public available pretrained T2I diffusion model, we experiment our approach on customization T2I generation tasks and show that our approach can capture visual attributes into prompt distribution and can generate diverse in-distribution images that follows text-guided editing.
- Further experiments show that our method is flexible in terms of diversity or mixing and easy to be adapted to other text-guided generation tasks.
- We further quantitatively demonstrate the effectiveness of our approach using synthetic image dataset generation tasks as a proxy and also through automatic evaluation metrics and human evaluation.

## 2 RELATED WORKS

### 2.1 TEXT-TO-IMAGE DIFFUSION MODELS

Diffusion models (Ho et al., 2020; Dhariwal & Nichol, 2021; Sohl-Dickstein et al., 2015) have achieved great success in various image generation tasks. State-of-the-art T2I models such as Imagen (Saharia et al., 2022) and DALL·E 2 (Ramesh et al., 2022) trained on large scale data demonstrate remarkable synthesis quality and controllability. Latent Diffusion Models (Rombach et al., 2021) and its open-source implementation, Stable Diffusion (Rombach et al., 2021), have also become a prevailing family of generative models. In these T2I diffusion models, text is encoded into latent vectors by pretrained language encoders such as CLIP (Radford et al., 2021), and the denoising process is conditioned on latent vectors to achieve text-to-image synthesis. However, such models trained on large scale text-image pairs are not designed to generate personalized images such as images of one's pet dog, therefore only the text conditioning cannot provide fine-grained control over the generated images.

### 2.2 PERSONALIZED TEXT-TO-IMAGE GENERATION

Various approaches are proposed to better control the text-guided diffusion models and achieve personalization. Textual Inversion (Gal et al., 2022) proposed to search for a new token in the embedding space representing a visual concept via optimizing a word embedding vector. DreamBooth (Ruiz et al., 2022) fine-tunes all parameters of the model to associate a personalized subject into an rarely used token. Custom Diffusion (Kumari et al., 2023) employs that fine-tuning method but only fine-tune cross-attention layers to reduce the training time, with the ability to learn multiple concepts jointly. Subsequent works (Voynov et al., 2023; Wei et al., 2023; Shi et al., 2023a; Alaluf et al., 2023) mainly follow the ideas from these works and focus on solving their drawbacks.

### 2.3 PROMPT LEARNING

Prompt learning is a popular method in natural language processing (NLP). The main idea is to transfer various downstream NLP tasks to masked language modeling problems via adopting proper prompt templates (Brown et al., 2020; Lester et al., 2021; Li & Liang, 2021; Radford et al., 2019) instead of fine-tuning the pretrained language model. Searching for the appropriate prompts is the key of this method. Prompt engineering (Brown et al., 2020; Radford et al., 2019) adopts carefully-designed discrete (hard) prompts crafting by human, while prompt tuning (Lester et al., 2021; Li & Liang, 2021) automatically searches for the desired prompts in the embedding space via learning continuous (soft) prompts. The great success of NLP inspires computer vision researchers and prompt engineering is explored in pretrained vision-language models such as CLIP (Radford et al., 2021) and ALIGN (Jia et al., 2021). CoOp (Zhou et al., 2022b) applies the idea of prompt tuning in vision-language tasks, which learns a continuous prompt via minimizing the classification loss of the downstream tasks. ProDA (Lu et al., 2022) learns a distribution of diverse prompts to capture various representations of a visual concept instead of a single prompt in CoOp (Zhou et al., 2022b), which achieves better generalization.

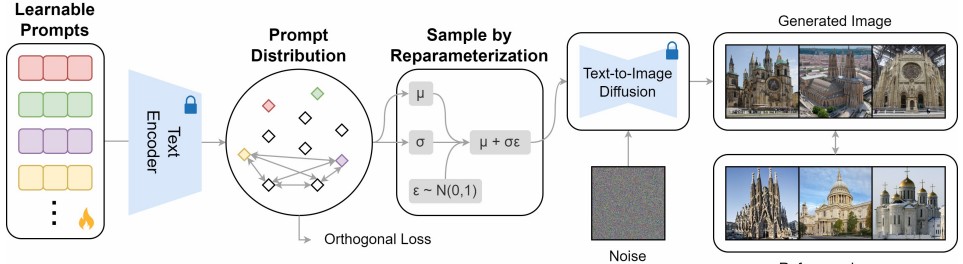

Figure 2: Overview of DreamDistribution for learning a prompt distribution. We keep a set of $K$ learnable soft prompts and model a distribution of them at the CLIP text encoder feature space. Only prompts are learnable, CLIP encoder and the T2I diffusion model are all fixed. We use reparameterization to sample from the prompt distribution and update the prompts through backpropagation. The training objective is to make the generated images aligns with the references. An additional orthogonal loss is incorporated to promote differentiation among learnable prompts. For inference, we sample from the prompt distribution at text feature space to guide the pretrained T2I generation.

Most relevant to our work are Textual Inversion (Gal et al., 2022) and ProDA (Lu et al., 2022). Textual Inversion learns a fixed token embedding associated with a pseudo-word. Ours learns a distribution of prompts in the CLIP feature space like ProDA, allowing for learning the visual concept with diverse representations and capturing the details for reconstructions and plausible synthesis.

## 3 METHOD

Given a set of images with some common visual attributes (*e.g.* same category, similar style), our goal is to capture the visual commonalities and variations and model by a prompt distribution in the text feature space. The commonalities among reference images may be challenging to articulate with natural language prompts. We can thus sample prompts from the distribution to guide T2I model to generate diverse unseen images while at the same time following the common traits distribution. The inherent characteristics of the learned prompts are compatible with natural language instructions and other pretrained text-guided generation models.

### 3.1 TEXT-TO-IMAGE DIFFUSION

Text-to-image diffusion models are a class of generative models that learns image or image latent distribution by gradually denoising a noise sampled from Gaussian distribution. Specifically, given a natural language text prompt, a tokenizer followed by a text embedding layer maps the input text to a sequence of embedding vectors $\mathbf{p}$. A text encoder converts the text embedding into text features $\mathbf{c} = \mathcal{E}(\mathbf{p})$ used for conditioning the generation process. An initial noise $\boldsymbol{\epsilon}$ is sampled from $\mathcal{N}(\mathbf{0}, \mathbf{I})$, and the denoising model $\boldsymbol{\epsilon}_\theta$ predicts the noise added to a noisy version of image of image latent $\mathbf{x}$. The denoising model $\boldsymbol{\epsilon}_\theta$ is optimized using the objective:

$$\mathcal{L} = \mathbb{E}_{\mathbf{x}, \mathbf{c}, \boldsymbol{\epsilon}, t} \left[ \| \boldsymbol{\epsilon} - \boldsymbol{\epsilon}_\theta \left( \mathbf{x}_t, \mathbf{c}, t \right) \|_2^2 \right] \tag{1}$$

where $\mathbf{x}$ is the ground-truth image or image latent obtained from a learned autoencoder, $\mathbf{x}_t$ is the noisy version of $\mathbf{x}$ at time-step $t$, and $\boldsymbol{\epsilon} \sim \mathcal{N}(\mathbf{0}, \mathbf{I})$.

### 3.2 PROMPT TUNING

Our proposed method is grounded in the notion of prompt tuning, which aims to learn a soft continuous prompt on target task and is widely used in fine-tuning NLP models. (Lester et al., 2021; Liu et al., 2023; Li & Liang, 2021; Hambardzumyan et al., 2021; Liu et al., 2021) Specifically, for a pretrained model that takes natural language prompt as input, we can formulate a prompt with continuous learnable token embeddings $\mathcal{P} = [\text{PREFIX}] \, \mathbf{V} \, [\text{SUFFIX}] \in \mathbb{R}^{L \times d}$, where [PREFIX] and [SUFFIX] are word embeddings of natural language prefix and suffix if needed, and $L$ represents the prompt length or the total number of tokens, and $d$ represent the dimension of word embeddings. $\mathbf{V} = [\mathbf{v}]_1 \ldots [\mathbf{v}]_M \in \mathbb{R}^{M \times d}$ represents a sequence of $M$ learnable token embedding vectors with same dimension as word embeddings. During fine-tuning, the parameters of the pretrained generation model remain fixed, and only the learnable token embeddings $\mathbf{V}$ are updated

through direct optimization employing the corresponding loss function backpropagated through generator $\boldsymbol{\epsilon}_\theta$ and text encoder $\mathcal{E}$. Formally, prompt tuning aims to find optimized embedding vectors $\mathbf{V}^* = \arg\max_{\mathbf{V}} P(Y \mid \mathcal{P}, X)$, where $X$ and $Y$ are input data and output label, respectively.

Prior works have shown the efficacy of adopting prompt tuning techniques on vision-language models for image classification tasks (Zhou et al., 2022b; Jia et al., 2022; Zhou et al., 2022a). Gal *et al.* (Gal et al., 2022) adopts similar prompt tuning methods that enable personalized generation. However, the limitation of this approach lies in its constraint to personalize only one particular concept, such as a specific dog, as it employs a fixed token embedding for concept encoding.

### 3.3 LEARNING PROMPT DISTRIBUTION

We aim to model more general commonalities and variations presented in the reference image set and generate diverse images of new instances that visually align, therefore we propose to model a learnable distribution of prompts for the reference images. Inspired by Lu *et al.* (Lu et al., 2022), which proposed to estimate a distribution of prompt for image classification tasks, we propose to model a distribution of learnable prompts over a sequence of $M$ token embeddings to capture the distribution of visual attributes on T2I generation task leveraging diffusion model.

Our methods builds on Stable Diffusion (Rombach et al., 2021), where a pretraind CLIP (Radford et al., 2021) text encoder is used for obtaining text feature of the prompt. Due to the contrastive training objective of CLIP, features of texts that have similar semantic meaning have high cosine similarity and therefore close to each other in CLIP feature space (Radford et al., 2021). Lu *et al.* (Lu et al., 2022) and some other previous works (Chen et al., 2022; Ma et al., 2023) have also shown that for text prompts that describe images of the same category, the CLIP text feature $\mathbf{c}$ output from pretrained CLIP text encoder are adjacent to each other in a cluster. Therefore, it is sufficient to model a Gaussian distribution of $\mathbf{c}$ that describes images of same category or with shared attributes without extra restriction. To do so, instead of keeping one learnable soft prompt to optimize during training, we maintain a set of $K$ learnable prompts $\mathcal{P}^K = \{\mathcal{P}_k = [\text{PREFIX}] \, \mathbf{V}_k \, [\text{SUFFIX}]\}_{k=1}^{K}$ corresponds to a set of similar reference images. Our goal is to optimize the set of learnable token embeddings $\{\mathbf{V}_k\}_{k=1}^{K}$. With $K$ learnable prompts, we can estimate the mean $\boldsymbol{\mu}_\mathbf{c} = \boldsymbol{\mu}\left(\mathcal{E}\left(\mathcal{P}^K\right)\right) \in \mathbb{R}^{L \times d_\mathcal{E}}$ and standard deviation $\boldsymbol{\sigma}_\mathbf{c} = \boldsymbol{\sigma}\left(\mathcal{E}\left(\mathcal{P}^K\right)\right) \in \mathbb{R}^{L \times d_\mathcal{E}}$ at $\mathcal{E}$ text encoder space, where $d_\mathcal{E}$ is the feature dimension of text encoder space.

Applying to the training objective of T2I diffusion model, eq. (1) becomes:

$$\mathcal{L}\left(\mathcal{P}^K\right) = \mathbb{E}_{\mathbf{x}, \tilde{\mathbf{c}}, \boldsymbol{\epsilon}\, t} \left[\|\boldsymbol{\epsilon} - \boldsymbol{\epsilon}_\theta\left(\mathbf{x}_t, \tilde{\mathbf{c}}, t\right)\|_2^2\right] \tag{2}$$

where $\tilde{\mathbf{c}} \sim \mathcal{N}\left(\boldsymbol{\mu}_\mathbf{c}, \boldsymbol{\sigma}_\mathbf{c}^2\right)$ and $\boldsymbol{\epsilon} \sim \mathcal{N}(\mathbf{0}, \mathbf{I})$ is the sampled Gaussian noise added to the image or image latent. However, sampling $\tilde{\mathbf{c}}$ from a distribution makes it not differentiable for optimization, therefore we apply the reparameterization trick similar to that used in VAE (Kingma & Welling, 2013). Formally, since $\tilde{\mathbf{c}} \sim \mathcal{N}\left(\boldsymbol{\mu}_\mathbf{c}, \boldsymbol{\sigma}_\mathbf{c}^2\right)$, we can rewrite the optimization objective eq. (2) as:

$$\mathcal{L}\left(\mathcal{P}^K\right) = \mathbb{E}_{\mathbf{x}, \boldsymbol{\omega}, \boldsymbol{\epsilon}, t} \left[\|\boldsymbol{\epsilon} - \boldsymbol{\epsilon}_\theta\left(\mathbf{x}_t, \boldsymbol{\mu}_\mathbf{c} + \boldsymbol{\omega}\boldsymbol{\sigma}_\mathbf{c}, t\right)\|_2^2\right] \tag{3}$$

where $\boldsymbol{\omega} \sim \mathcal{N}\left(\mathbf{0}, \mathbf{I}\right)$ has the same dimension as $\boldsymbol{\mu}_\mathbf{c}$ and $\boldsymbol{\sigma}_\mathbf{c}$. Since the exact computation of $\mathcal{L}\left(\mathcal{P}^K\right)$ is intractable, we use a Monte Carlo approach to sample $\boldsymbol{\omega}$ for $S$ times to approximate the expected value for optimization:

$$\mathcal{L}\left(\mathcal{P}^K\right) = \frac{1}{S} \sum_{s=1}^{S} \|\boldsymbol{\epsilon} - \boldsymbol{\epsilon}_\theta\left(\mathbf{x}_t, \boldsymbol{\mu}_\mathbf{c} + \boldsymbol{\omega}_s \boldsymbol{\sigma}_\mathbf{c}, t\right)\|_2^2 \tag{4}$$

In order to avoid the scenario wherein multiple prompt features converge to a same vector, which will result in a non-representative low-variance distribution, we apply a similar orthogonal loss proposed in (Lu et al., 2022) to penalize on the cosine similarity and encourage orthogonality between each pair of prompts:

$$\mathcal{L}_{\text{ortho}} = \frac{1}{K(K-1)} \sum_{i=1}^{K} \sum_{j=i+1}^{K} |\langle \mathcal{E}(\mathcal{P}_i), \mathcal{E}(\mathcal{P}_j) \rangle| \tag{5}$$

where $\langle \cdot, \cdot \rangle$ is cosine similarity between a pair of vectors. The total loss is therefore:

$$\mathcal{L} = \mathcal{L}(\mathcal{P}^K) + \lambda \mathcal{L}_{\text{ortho}} \tag{6}$$

where $\lambda$ is a hyperparameter.

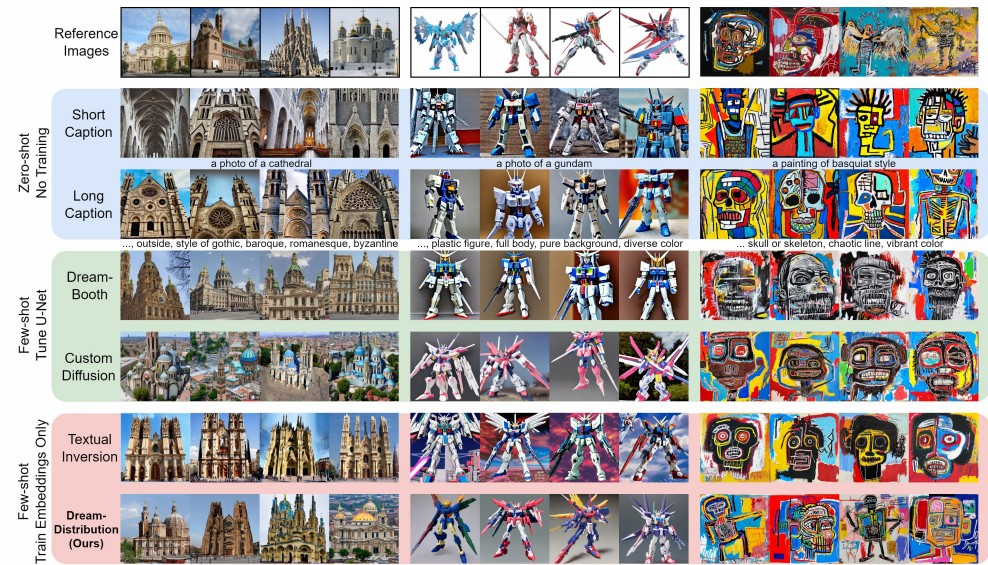

Figure 3: Comparison of results with existing methods. Given a set of training images (typically 5-20, we only show 4 here), we compare generation results with other existing methods. We use Stable Diffusion version 2.1 for all methods. As can be seen on the bottom row, our method is able to generate more diverse and coherent images (also quantitatively analyzed by automatic and human evaluation in §4.2).

**Implementation Details**    In all experiments, we use Stable Diffusion 2.1 (Rombach et al., 2021) and keep all the default hyperparameters. We use $S = 4$ and $\lambda = 5 \times 10^{-3}$. We use $K = 32$ prompts in all personalized generation experiments, and $K = 10$ prompts to reduce computation in synthetic dataset experiments. We use 1,500 steps with constant learning rate of $10^{-3}$.

## 4    EXPERIMENTS

In this section, we demonstrate several experiments and applications of our approach and show visual results of generated images. We show the ability of our approach to capture a distribution of reference images and generate in-distribution novel images in §4.1. We present additional quantitative results including automatic evaluation and user studies in §4.2. We also show the flexibility and effects of manipulating and text-guide editing learned prompt distribution in §4.3. We further highlight easy application of our learned prompt distribution to other text-based generation tasks using text-to-3D as an example in §4.4. Finally in §4.5 we present experiments that show the effectiveness of our approach in generating synthetic training dataset.

### 4.1    DIVERSE PERSONALIZED GENERATION

We first demonstrate the ability of our approach to generate images that preserve general visual features shown in training set and at the same time generate new images with high diversity. Given a diverse set of few training images (typically 5-20) that are not easily describable in texts and at the same time share some similar visual attributes, we can generate diverse in-distribution images by simply sampling from the learned distribution as the input prompt text embedding to T2I diffusion model. Our learned prompt distribution can be therefore treated as a distribution of descriptions corresponding to the set of training images.

**Baselines.**    We compare with popular instance-level personalization methods including **Textual Inversion** (Gal et al., 2022), **DreamBooth** (Ruiz et al., 2022), **Custom Diffusion** (Kumari et al., 2023). We also evaluate against **Short Caption** that uses a short description as text prompt, and **Long Caption** that uses a longer text caption with detailed descriptions. These comparisons emphasize our method's ability to take care of both similarity and diversity referencing the training images. We use the same pretrained Stable Diffusion version 2.1 with default hyperparameters provided in baseline works. We use $M = 8$ context vectors without adding any prefix or suffix texts in either training or inference process for DreamDistribution.

| Method | CLIP-I↑ | CLIP-T↑ | DINO↑ | Density↑ | Coverage↑ | FID↓ |
|---|---|---|---|---|---|---|
| DreamBooth | 0.79 | 0.32 | 0.46 | 0.91 | 0.74 | 234.90 |
| Textual Inversion | 0.83 | 0.27 | 0.48 | 1.28 | 0.82 | 224.23 |
| Custom Diffusion | 0.80 | **0.33** | 0.46 | 1.45 | 0.87 | 236.61 |
| Ours | **0.84** | 0.29 | **0.50** | **1.59** | **0.93** | **215.15** |

(a) Our method achieves the best number on most automatic metrics. Mean value of the metrics across 12 scenarios are reported.

(b) Human Evaluation on image diversity (§4.2) aligns with automatic evaluation (Fig. 4a). Our method shows significantly greater diversity, which may explain why it can train better image classifiers in Tab. 1.

Figure 4: Quantitative and human evaluation.

**Results.** Fig. 3 shows visualized comparison with baselines. In general, both short and long text prompting methods fail to generate results that visually follow the reference images since there is no training involved and the image details are hard to describe in language. Images generated using baseline methods generally show limited variation or inconsistent visual attributes in all examples. All these methods try to associate different visual concepts with a fixed token, which does not provide any semantic variations itself. Although the denoising process enables some randomness, the training objective of associating various concepts with a fixed token will either fail to capture a distribution due to non-convergence, leading to underfitting to generic image category information, or overfits to a visual combination of the training images. By modeling multiple concepts using multiple prompts and optimizing the prompt distribution, our proposed method is able to produce substantial variations of style and view points, for example, following the reference images in the cathedral example (first column). Ours method can also model the texture and background information and generate new instance with significant variations in color and pose following the reference images of the Gundam example (second column), as well as patterns, lines, style as a whole and generate novel artistic creations as shown in the Basquiat's painting example (third column). In all, DreamDistribution is able to produce substantial variations on style, viewpoints, pose, layout, etc., with appropriate visual attributes following the reference images.

### 4.2 GENERATION QUALITY AND DIVERSITY EVALUATION

We quantitatively assess our methods in terms of diversity and quality, and further use synthetic ImageNet classification performance as a proxy in §4.5. We also conduct human evaluation study on the same generation result to compare with baseline methods.

**Datasets** Different from existing datasets such as DreamBooth Dataset (Ruiz et al., 2022), which focus on same-instance personalization, we construct a dataset that consists of different instances in a same category set. Our dataset consists of 12 diverse image scenarios including photos of real objects in large and small scales, works of famous artists, and illustrations with prominent styles, sourced from illustrators from online communities. Each scenario consists of about 20 real reference images. A comprehensive list and visualization is shown in the appendix. We train our method and baselines including DreamBooth, Textual Inversion, and Custom Diffusion using the default settings provided. For our approach we use $M = 4$ learnable context with no text prefix or suffix in both training and generating stages. For baselines, we generate image using the prompt "*a photo/drawing of S\**" depending on the nature of the reference set, where $S\*$ is the learned token(s). We generate 40 images per methods per scenario, resulting in a total of 1,920 images in the evaluation set.

**Automatic Metrics** We evaluate the generative images on established automatic evaluation metrics that measure the diversity of synthetic images and the similarity between real and synthetic images. Following prior works (Heusel et al., 2017; Zhang et al., 2018; Naeem et al., 2020; Ruiz et al., 2022), in Fig. 4a we evaluate image quality using **CLIP-I** and **DINO** (Ruiz et al., 2022) that measures average pairwise cosine similarity between CLIP (Radford et al., 2021) and DINOv1 (Caron et al., 2021) embeddings, and **FID** (Heusel et al., 2017) that measures the distance between the distribution of generated images and the distribution of real images via InceptionV3 (Szegedy et al., 2016). Our method achieves the best quality across all three quality measurements, suggesting

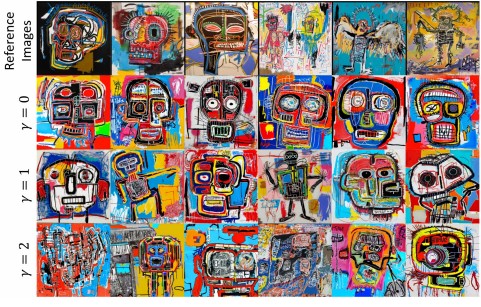 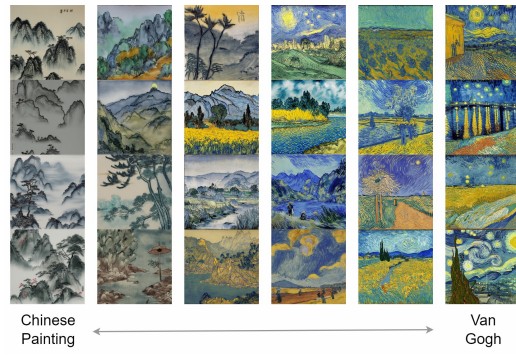

Chinese Painting ⟷ Van Gogh

(a) Effect of scaling the variance of a learned prompt distribution. Image diversity increases as the scaling factor $\gamma$ increases.

(b) Composition of two prompt distributions through interpolation. Mixing ratio changes linearly from left to right. Middle columns show mixtures of two styles.

Figure 5: Scaling variance and composition of prompt distributions.

that our method is capable of creating more high-quality images that fulfill the prompt requirement. **CLIP-T** measures the image-text alignment in CLIP space for text-editing generation. We use the 25 object prompts provided in (Ruiz et al., 2022) and generate 4 images per prompt per scenario, resulting 1,200 text-editing images. Our method may not show best performance due to the random nature of distribution but still gives comparable result. Additionally, we report **Density** and **Coverage** (Naeem et al., 2020) in Fig. 4a. Density measures samples in regions where real samples are densely packed, while coverage calculates fraction of real samples whose neighbourhoods contain at least one generated sample. Both metrics are calculated with DINOv2 (Oquab et al., 2023). Our method achieves the best coverage and diversity across the board.

**Human Evaluation**  Admittedly, automatic evaluation does not fully capture the richness perceived by human observers. We further investigate if Fig. 4a correlates with human perception via conducting human evaluation based on the evaluation dataset. We assign 20 independent annotators. For each of the 12 reference sets, annotators are asked to choose the most preferable set of generated images based on their perceived similarity with the reference set and the diversity within the generated set. The methods are anonymized so annotators are unaware of which generated set corresponds to which method. We collect a total of 240 samples and count the frequency of preferences. Fig. 4b demonstrates that our generated images exhibit superior diversity compared to three baseline models, reinforcing our intuition that by learning distribution we are able to generate diverse images with coherent content and visual attributes presented in the reference image.

### 4.3 Controllability of Prompt Distribution

Since our learned prompt distribution is in the CLIP text feature space, it is natural to manipulate the learned distribution based on the property of CLIP text feature space. We show several interesting distribution manipulation methods, including text-guided editing, scaling the variance for diversity control, interpolation between multiple distributions.

**Text-guided Editing**  Similar to existing personalization methods (Ruiz et al., 2022; Gal et al., 2022; Kumari et al., 2023), our learned distribution preserves the flexibility of text-guided editing . As shown in Fig. 1 and Fig. 6, we are able to generate diverse in-distribution Gundam figures that follows the style of reference images but with different pose, style, context, using user provided text-guidance at inference time. With a set of learned prompt, we concatenate them with the same text prefix and/or suffix to fit a new distribution at the CLIP text feature space to enable text-guided editing of a prompt distribution. Application includes but not limited to, generating objects of interests in a different background or context, transferring style using text, and controlling the pose, viewpoints, layout, of objects of interests.

**Scaling Variance for Diversity Control**  Once a prompt distribution is learned, we can easily control the diversity of generated images by changing the variance or standard deviation of the learned distribution. We show an example of the effect of multiplying different scale factors $\gamma$ to the variance of a learned prompt distribution in Fig. 5a. When $\gamma = 0$, the generated images show very similar patterns following some of the reference images. As $\gamma$ increases, more different layouts emerge, and when we further scale the variance for $\gamma = 2$, the generated images become more diverse with significant randomness.

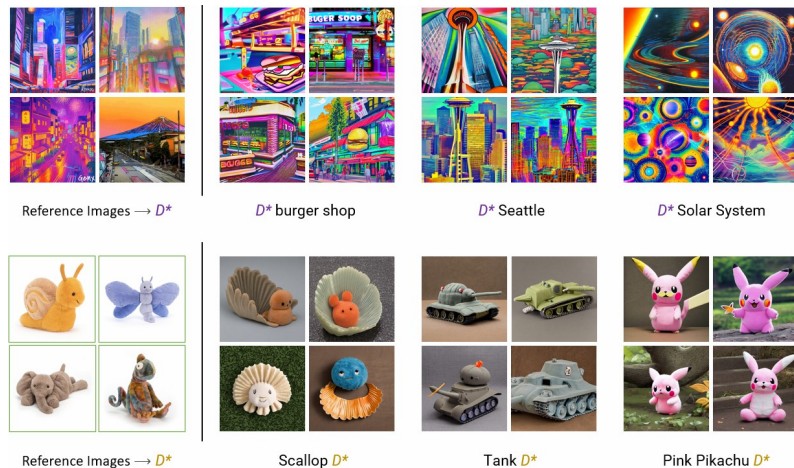

Figure 6: Results on text-editability of our methods. Left column shows samples of reference images, right columns are generated results with corresponding prompts.

**Composition of Distributions**   Given multiple prompt distributions in CLIP feature space, we can composite distributions by finding a linearly interpolated distribution between them. This distribution in the CLIP feature space should represent a text with semantic meaning that is a weighted mixture of the given prompt distributions, thereby showing a mixture of visual attributes in the generated images. We naively use a weighted sum of the distributions to interpolate between distributions:

$$\boldsymbol{\mu}_{\mathbf{c}}^* = \sum_{i=1}^{N} \alpha_i \boldsymbol{\mu}_{\mathbf{c}_i}, \quad \boldsymbol{\sigma}_{\mathbf{c}}^* = \sum_{i=1}^{N} \sqrt{\alpha_i} \boldsymbol{\sigma}_{\mathbf{c}_i} \tag{7}$$

where $\boldsymbol{\mu}_{\mathbf{c}}^*$ and $\boldsymbol{\sigma}_{\mathbf{c}}^*$ are mean and standard deviations of the interpolated distribution, and $\alpha_i$ is the weight of $i$-th prompt distribution with mean and standard deviation $\boldsymbol{\mu}_{\mathbf{c}_i}$ and $\boldsymbol{\sigma}_{\mathbf{c}_i}$ respectively, and $\sum_{i=1}^{N} \alpha_i = 1$ are weight parameters. We show an example of mixing Chinese paintings and Van Gogh paintings in Fig. 5b. From the left column to right, we adjust the mixing ratio to increase the weight of Van Gogh and decrease the weight of Chinese painting.

### 4.4 APPLYING TO TEXT-TO-3D GENERATION

Our learned distribution can be flexibly applied to other text-driven tasks, as long as the generation pipeline uses the same pretrained text encoder as the text feature extractor. In this section, we highlight and demonstrate the flexibility of our method by using a prompt distribution trained on T2I diffusion for text-to-3D task. We use MVDream (Shi et al., 2023b), a state-of-the-art text-to-3D model that train a NeRF (Mildenhall et al., 2021) and render a 3D asset following a text prompt, which in our case is a prompt sampled from prompt distribution. As shown in Fig. 1 and Fig. 7a, although MVDream incorporates some extra prior in its modified multi-view diffusion model that leads to reduced diversity, our prompt distribution can still generate 3D assets with significant variation in design details. Moreover, as shown in Fig. 7b, the pipeline possesses text-guided editing capabilities akin to those of DreamBooth3D (Raj et al., 2023), yet it can generate instances that exhibit more diverse appearances.

### 4.5 APPLYING TO SYNTHETIC DATASET GENERATION

Our proposed method can be effectively used in generating synthetic image classification datasets. By giving several dozens to hundreds of images that correspond to a class in a classification dataset, our method can capture and encode distributions of the dataset images into the learnable prompt distributions, and thereby generate diverse training images similar to the training set.

We generate "synthetic copy" (Ge et al., 2022a;b; Sariyildiz et al., 2022; Ge et al., 2023) of ImageNet (Russakovsky et al., 2015) via DreamDistribution using Stable Diffusion version 2.1 with default hyperparameters. Due to the large size of ImageNet-1K, we follow previous works (Sariyildiz et al., 2022) to mainly experiment on **ImageNet-100** (IN) (Tian et al., 2020), a 100-class subset. For each class, we generate 2,000 synthetic images and use CLIP (Radford et al., 2021) to select top

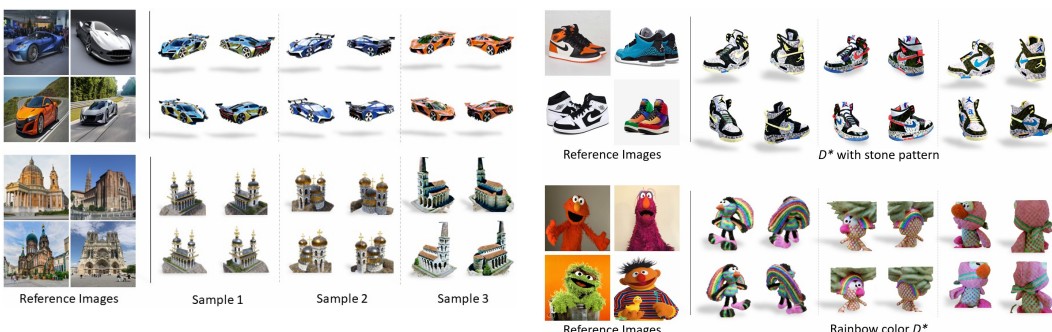

(a) 3D generation results by learning a prompt distribution over reference images and then inference (without extra texts).

(b) 3D generation results by learning a prompt distribution over reference images and then inference with text-guided editing.

Figure 7: Results of our method applied to 3D generation using MVDream (Shi et al., 2023b).

| Method | IN | | IN-V2 | | IN-S | | IN-R | | IN-A | | CLIP-I↑ | DINO↑ | Density↑ | Coverage↑ | FID↓ |
|---|---|---|---|---|---|---|---|---|---|---|---|---|---|---|---|
| | Top1 | Top5 | Top1 | Top5 | Top1 | Top5 | Top1 | Top5 | Top1 | Top5 | | | | | |
| Real | 88.0 | 96.7 | 85.1 | 94.9 | 45.1 | 63.9 | 66.1 | 85.2 | 26.7 | 65.8 | - | - | - | - | - |
| Class Names | 45.5 | 70.0 | 46.2 | 72.5 | 24.1 | 43.3 | 53.6 | 75.8 | 8.1 | 38.8 | 0.75 | 0.43 | 0.71 | 0.41 | 149.87 |
| CLIP Prompts (Radford et al., 2021) | 45.6 | 69.2 | 46.1 | 69.6 | **36.2** | **60.1** | 58.8 | 81.1 | 12.2 | 45.7 | 0.71 | 0.37 | 0.63 | 0.40 | 166.42 |
| ImageNet-SD (Sariyildiz et al., 2022) | 55.4 | 77.5 | 55.8 | 77.5 | 29.4 | 49.0 | **59.8** | **80.0** | 15.9 | 49.4 | 0.72 | 0.38 | 0.54 | 0.36 | 173.58 |
| Textual Inversion (Gal et al., 2022) | 17.3 | 35.7 | 17.7 | 35.3 | 4.84 | 13.2 | 19.2 | 46.6 | 8.2 | 33.2 | 0.71 | 0.35 | 0.43 | 0.21 | 214.14 |
| DreamBooth (Ruiz et al., 2022) | 44.1 | 67.4 | 45.2 | 67.7 | 11.4 | 28.1 | 33.4 | 59.2 | 8.8 | 38.8 | 0.77 | **0.50** | **0.89** | 0.51 | 139.49 |
| Custom Diffusion (Kumari et al., 2023) | 34.0 | 57.6 | 36.0 | 59.7 | 6.1 | 15.7 | 24.7 | 50.1 | 10.1 | 40.3 | 0.75 | 0.47 | 0.84 | 0.42 | 141.53 |
| DreamDistribution (Ours) | **56.7** | **79.8** | **62.0** | **81.5** | 30.4 | 50.9 | 54.7 | 78.4 | **18.8** | **55.0** | **0.78** | 0.48 | **0.89** | **0.56** | **131.60** |

Table 1: Classification accuracy and other metrics on different real test sets by training a classifier on synthetic ImageNet (IN) generated by different methods. The rows with gray background are non-training methods. When training on images generated from our method, the resulting classifier performs better on most test sets, indicating that the images synthesized by our method allowed the classifier to learn those object categories better.

1,300 images with highest cosine similarity to the embedding vector of the corresponding class name, resulting the same total number of images as real ImageNet training set. We compare with non-training baselines: **Real** uses the real ImageNet training set, **Class Names** and **CLIP Prompts** generate images by feeding Stable Diffusion class name of each class or with 80 diverse text prompts provided by CLIP, *e.g.* "a drawing of $c$", where $c$ is the class name. **ImageNet-SD** (Sariyildiz et al., 2022) generates images using prompts in the form of "$c$, $h_c$ inside $b$", where $c$ represents the class name, $h_c$ represents the hypernym (WordNet parent class name) of the class, and $b$ is a random background description from the Places365 dataset (Zhou et al., 2017). We also experiment with other aforementioned personalization baselines. For personalization baselines and our method, we use same random 100 images per class to train the generation pipeline. Then we train a ResNet-50 (He et al., 2016) classifier on *synthetic images only* for 300 epochs using 0.2 alpha for mixup augmentation (Zhang et al., 2017) and auto augment v0 via `timm` (Wightman, 2019).

To analyze generalizability, we also evaluate the trained model on validation set of ImageNet variants including **ImageNetV2** (IN-V2) (Recht et al., 2019), **ImageNet-Sketch** (IN-S) (Wang et al., 2019), **ImageNet-R** (IN-R) (Hendrycks et al., 2021a), and **ImageNet-A** (IN-A) (Hendrycks et al., 2021b). Top-1 and top-5 accuracy is reported in Tab. 1. In all settings, the classifier is exclusively exposed to synthetic images, but images generated using our method shows the highest classification accuracy on ImageNet validation set. This is because DreamDistribution can generate a diverse set of high-quality images following training set distribution, while other prompt engineering methods cannot follow the real image distribution and tend to show limited diversity, resulting in performance degradation. We also achieve the best results on ImageNet-V2 and ImageNet-A. For the Sketch and Rendition variants, in contrast to our method, CLIP Prompts and ImageNet-SD offer specific prompts to generate images of other domains, which may account for our comparatively lower performance.

## 5 CONCLUSION

We introduced DreamDistribution, a distribution based prompt tuning method for personalizing T2I diffusion models to generate diverse in-distribution images following a small set of reference images. The key idea of our methods lies in modeling the commonalities and variations of visual attributes using a prompt distribution at text feature space. We show a variety of experiments and application that is enabled by our method.

## 6 ACKNOWLEDGMENT

This work was supported by the National Science Foundation (award 2318101), C-BRIC (one of six centers in JUMP, a Semiconductor Research Corporation (SRC) program sponsored by DARPA), the Army Research Office (W911NF2020053), and Amazon ML Fellowship. The authors affirm that the views expressed herein are solely their own, and do not represent the views of the United States government or any agency thereof.

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

## A    MORE IMPLEMENTATION DETAILS

In all experiments, we use Stable Diffusion 2.1, which is the latest version. We use the default parameters, including 7.5 guidance scale and 50 denoising steps. For all visual results, we generate images in $768 \times 768$ resolution, and for synthetic dataset experiments, we generate $256 \times 256$ images to save time and resources. We provide a pseudocode for learning a prompt distribution in Algorithm 1.

---

**Algorithm 1** Training prompt distribution

---

**Require:** Set of reference images $\mathcal{I} = \{\mathbf{x}_i\}_{i=1}^N$
**Require:** Set of learnable prompts $\mathcal{P}^K = \{\mathcal{P}_i\}_{i=1}^K$
**Require:** Text encoder $\mathcal{E}$, noise predictor $\epsilon_\theta$, hyperparameter $\lambda$
1: Random initialize all learnable embeddings $\mathbf{V}$ in $\mathcal{P}^K$
2: **for** image $\mathbf{x} \in \mathcal{I}$ **do**
3:     Sample time step $t$
4:     Sample noise $\epsilon \sim \mathcal{N}(0, \boldsymbol{I})$
5:     $\mathbf{x}_t \leftarrow \mathbf{x}$ with noise added based on $\epsilon$ and $t$
6:     Compute $\boldsymbol{\mu_c} = \frac{1}{K} \sum_{i=1}^K \mathcal{E}(\mathcal{P}_i)$
7:     Compute $\boldsymbol{\sigma_c^2} = \frac{1}{K} \sum_{i=1}^K (\mathcal{E}(\mathcal{P}_i) - \boldsymbol{\mu_c})^2$
8:     **for** $s \in [S]$ **do**
9:         Sample $\boldsymbol{\omega}_s \sim \mathcal{N}(0, \mathbf{I})$
10:         $\mathcal{L}_s(\mathcal{P}^K) = \|\epsilon - \epsilon_\theta(\mathbf{x}_t, \boldsymbol{\mu_c} + \boldsymbol{\omega}_s \boldsymbol{\sigma_c}, t)\|_2^2$
11:     **end for**
12:     $\mathcal{L}(\mathcal{P}^K) = \frac{1}{S} \sum_{s=1}^S \mathcal{L}_s(\mathcal{P}^K)$
13:     $\mathcal{L}_{\text{ortho}} = \frac{1}{K(K-1)} \sum_{i=1}^K \sum_{j=i+1}^K |\langle \mathcal{E}(\mathcal{P}_i), \mathcal{E}(\mathcal{P}_j) \rangle|$
14:     $\mathcal{L} = \mathcal{L}(\mathcal{P}^K) + \lambda \mathcal{L}_{\text{ortho}}$
15:     Update learnable embeddings $\mathbf{V}$ in $\mathcal{P}^K$ based on $\mathcal{L}$
16: **end for**

---

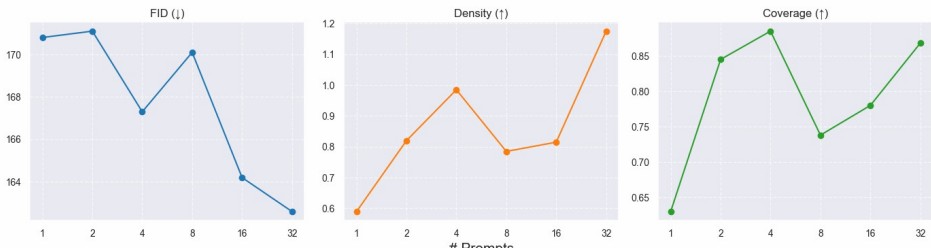

Figure 8: In general, with more prompts, the performance increases in terms of both quality and diversity.

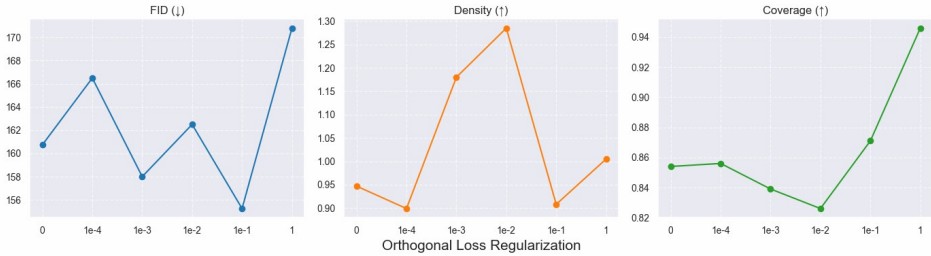

Figure 9: Choice of $\lambda$ value between $1 \times 10^{-4}$ and $1 \times 10^{-2}$ generally achieves good balance of quantitative metrics.

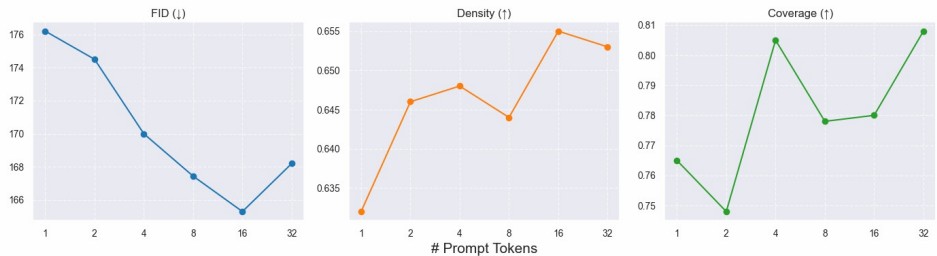

Figure 10: Similar to number of prompts, increasing number of prompt tokens also shows increasing image quality and diversity.

## B    EVALUATION SET

We show samples of reference images from our evaluation set in Fig. 11. Each row shows 8 samples reference images from the same set.

## C    ADDITIONAL RESULT

### C.1    DIVERSE IMAGE INSTANCE GENERATION

We show additional image generation results of our method in Fig. 12 using the reference images from our evaluation set. Each row is generated using reference images of the corresponding row in Fig. 11. We also show additional examples in Fig. 13 that demonstrate the ability of our method to capture visual similarities in reference images and generate diverse images, specifically on the attributes that are shown different in reference images.

### C.2    TEXT-GUIDED EDITING

We show more results on the ability of our method to generate diverse images with text-guided editing in Fig. 14 using different sets of reference images.

### C.3    SCALING VARIANCE FOR DIVERSITY CONTROL

We show more results on generating images from prompt distribution with scaled standard deviations in Fig. 15

### C.4    COMPOSITION OF DISTRIBUTION

In Fig. 16 we show more results on composition of two different learned prompt distributions with various weights.

### C.5    SAME INSTANCE PERSONALIZATION

Our method solves a more generalized task compared to same instance personalization, where reference images is one or more images of the same instance and the generation is expected to follow user provided text prompts of different contexts. Although reconstruction of the same instance is not our main focus, we still show that our method can achieve on par results on this task compared with baselines, using DreamBooth dataset (Ruiz et al., 2022) with identical prompts and evaluation settings. The quantitative results are show in Tab. 2.

### C.6    NAIVE ADAPTATION OF BASELINES

We compare our approach with a naive adaptation of baseline methods, where the training set is randomly divided into 4 subsets, and a generation model is trained on each subset. During inference,

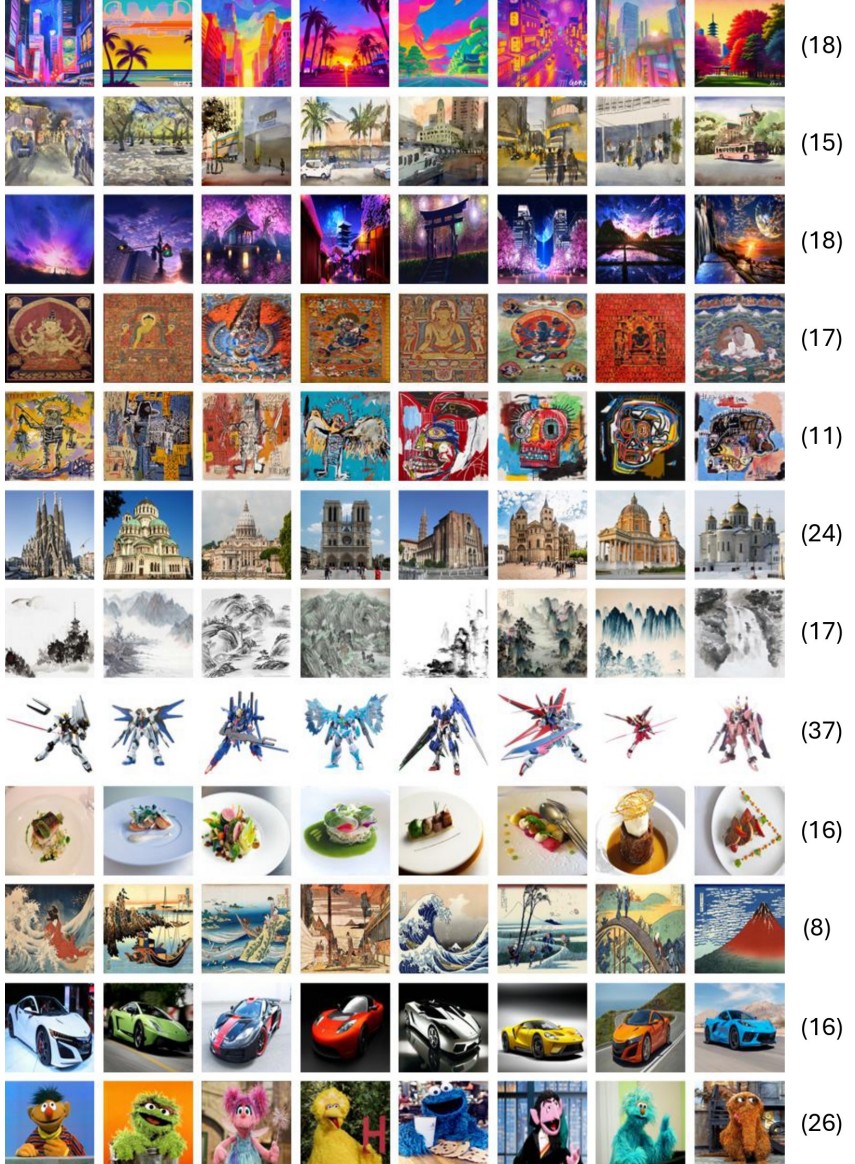

Figure 11: Samples of reference images from our evaluation set. Numbers on the right represent the number of images in each set.

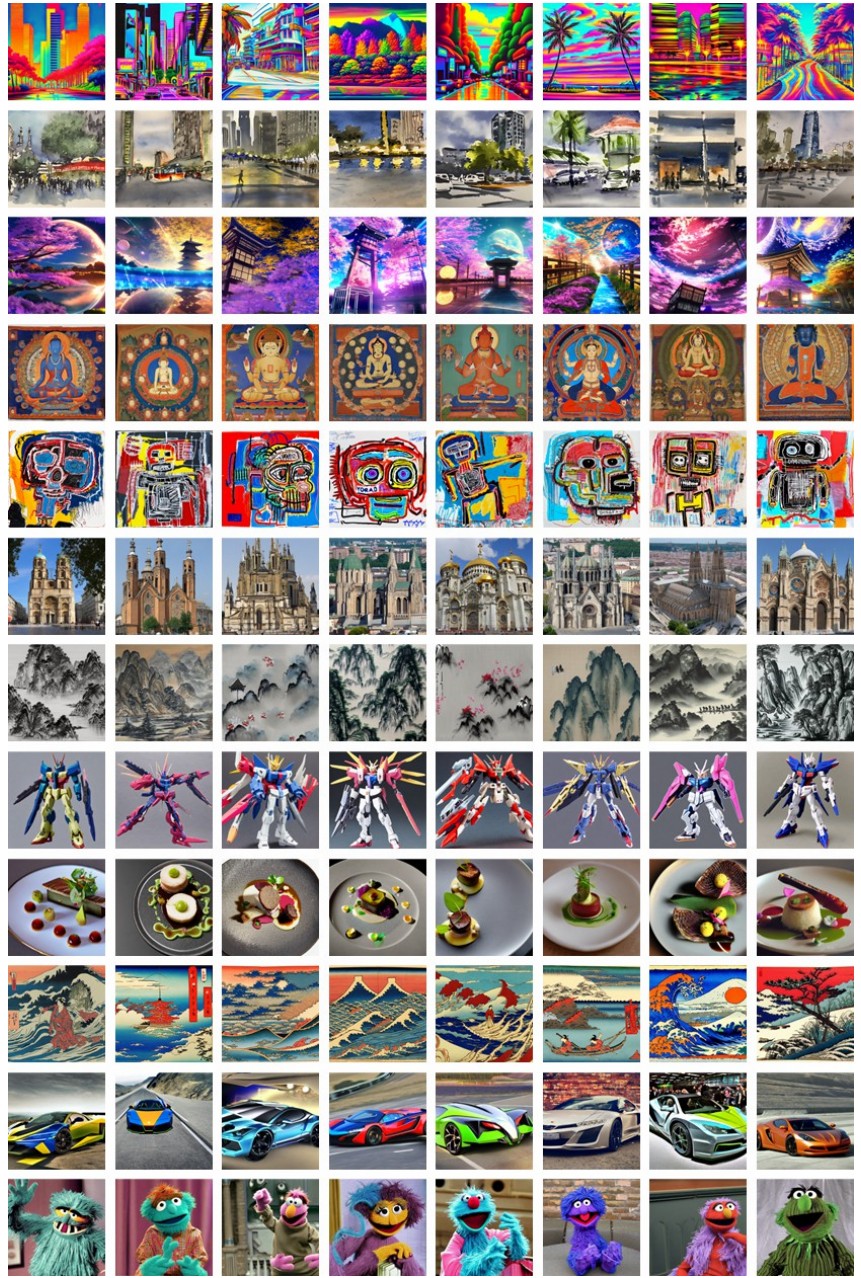

Figure 12: Samples of generated image results using reference images from the evaluation set. Each row is generated using reference images of the corresponding row in Fig. 11.

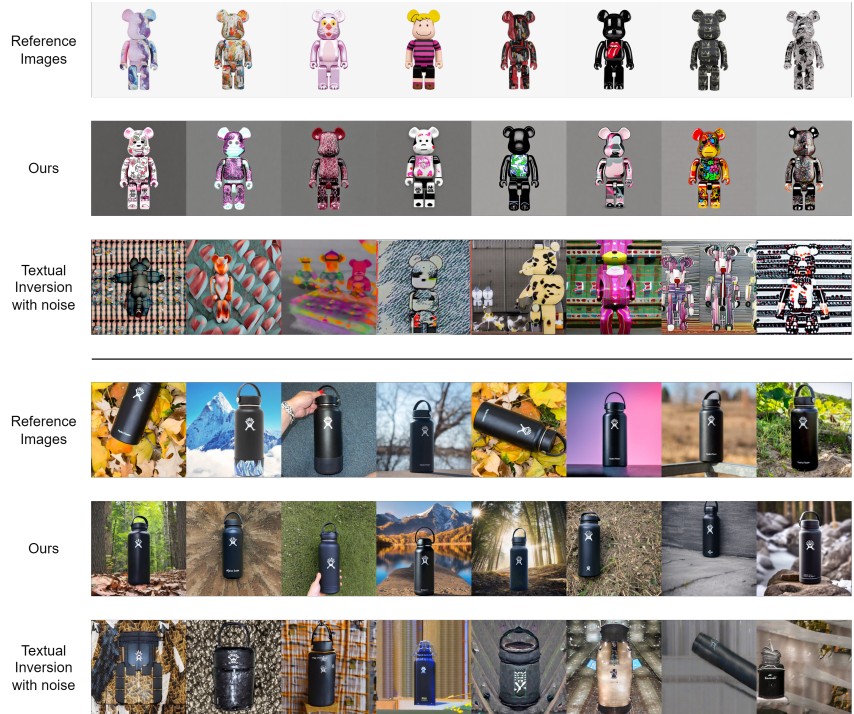

Figure 13: More results that shows the ability of our method to capture same visual attributes and generate diverse images specifically on the attributes that are shown different in reference images. The top example shows reference images with same shape outline but different foreground texture and pattern, and our method also generates the same shape with diverse new foreground patterns. The bottom example shows reference images of same object in different background, and our method generates the same object in diverse background as well. We additionally show comparison with a naive adaptation of Textual Inversion, where we add a Gaussian noise with 0.5 variance to the learned tokens for generation, and the result shows that although it may increase diversity due to introduction of extra randomness, it fails to capture the common attributes that should not be varied in both examples, such as constant shape and background in first example and color and appearance in second example.

| Method | CLIP-I↑ | CLIP-T↑ | DINO↑ |
|---|---|---|---|
| DreamBooth | 0.80 | 0.31 | 0.67 |
| Textual Inversion | 0.78 | 0.26 | 0.57 |
| Custom Diffusion | 0.80 | 0.36 | 0.54 |
| Ours | 0.80 | 0.31 | 0.59 |

Table 2: Experiment results of personalized generation on DreamBooth dataset.

the final output is a mixture of results from the models trained on these different subsets. Based on the evaluation results shown in Tab. 3, our method outperforms this naive adaptation of three baseline methods on all metrics. Moreover, this naive adaptation takes significantly more extra disk space since it requires the storage of multiple sets of model weights.

| Method | CLIP-I↑ | CLIP-T↑ | DINO↑ | Density↑ | Coverage↑ | FID↓ |
|---|---|---|---|---|---|---|
| DreamBooth | 0.80 | 0.26 | 0.44 | 1.00 | 0.83 | 232.13 |
| Textual Inversion | 0.80 | 0.25 | 0.44 | 0.78 | 0.66 | 243.93 |
| Custom Diffusion | 0.75 | 0.27 | 0.39 | 0.62 | 0.57 | 268.48 |
| Ours | **0.84** | **0.29** | **0.50** | **1.59** | **0.93** | **215.15** |

Table 3: Comparison of our methods with naive adaptations of baseline methods.

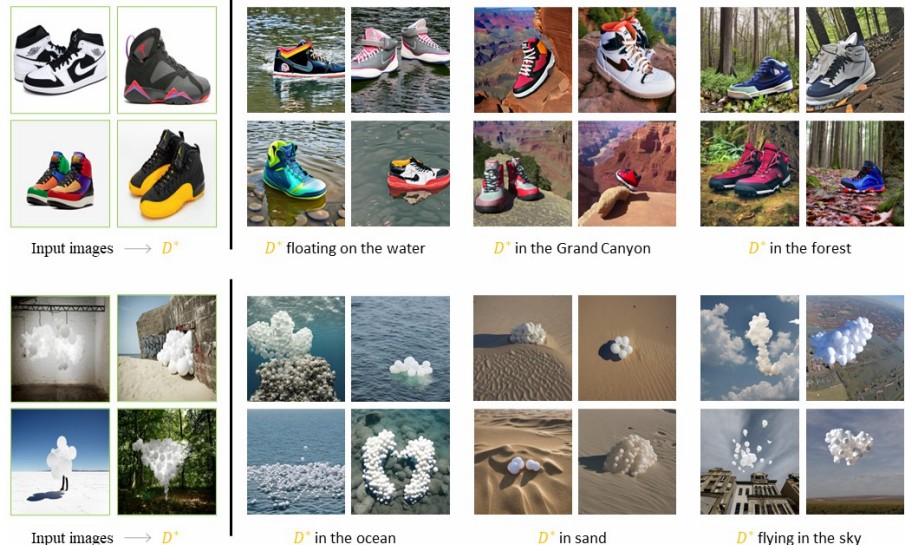

Figure 14: More results on text-editability of our methods. Left column shows samples of reference images to learn distribution $D^*$, right columns are generated results using prompts sampled from corresponding text-edited distribution.

## C.7 Exploring with Different Granularity of Concepts

We experiment on image sets at different class granularity to show that our method is capable of capturing distribution at different levels of concepts. Specifically, we composed 3 different reference image sets, forced from ImageNet images: French Bulldog, dogs, and quadruped animals, where dogs set includes images from different ImageNet dog species such as Bulldog, Saluki, Chihuahua, etc., and animals set includes images from different ImageNet quadruped classes, such as tiger, cat, bison, fox, etc. We run DreamDistribution on these image sets respectively. The result visualization is shown in Fig. 17. For French Bulldog, our method is able to generate different instances of French Bulldogs with different fur colors and appearances. For dogs reference set, our method is able to generate different dog species with mixed attributes. For quadrupeds reference set, our method can generate a variety of quadruped animals, including non-existing animals that show a mix of attributes from different animals. Quantitative analysis presented in Tab. 4 demonstrates that at the most granular level, such as French Bulldog, our method significantly outperforms the personalization baselines. However, at coarser-grained levels, such as Dogs or Quadrupeds, the advantage of our method diminishes. This could be attributed to the increased diversity of the reference images, which makes it challenging for our method to effectively capture a coherent or meaningful distribution, leading to a higher likelihood of sampling outliers from the learned prompt distribution during generation.

## C.8 Experiment with Small Reference Set

We conduct experiments on image sets with only four reference images to simulate scenarios where users are unable to provide a larger set for better distribution capture. Our method is compared against the personalization baselines used in previous sections. Visualizations of the generated results are shown in Fig. 18, which shows that our method can still generate results with more diversity and at the same time within the distribution of reference images.

## C.9 3D Generation Diversity

Our learned prompt distribution can extend to text-to-3D generation pipelines, such as MVDream Shi et al. (2023b), which utilize text-to-image models as their backbone, enabling more diverse 3D asset generation. As illustrated in Fig. 19, unlike the Textual Inversion baseline, which relies on a

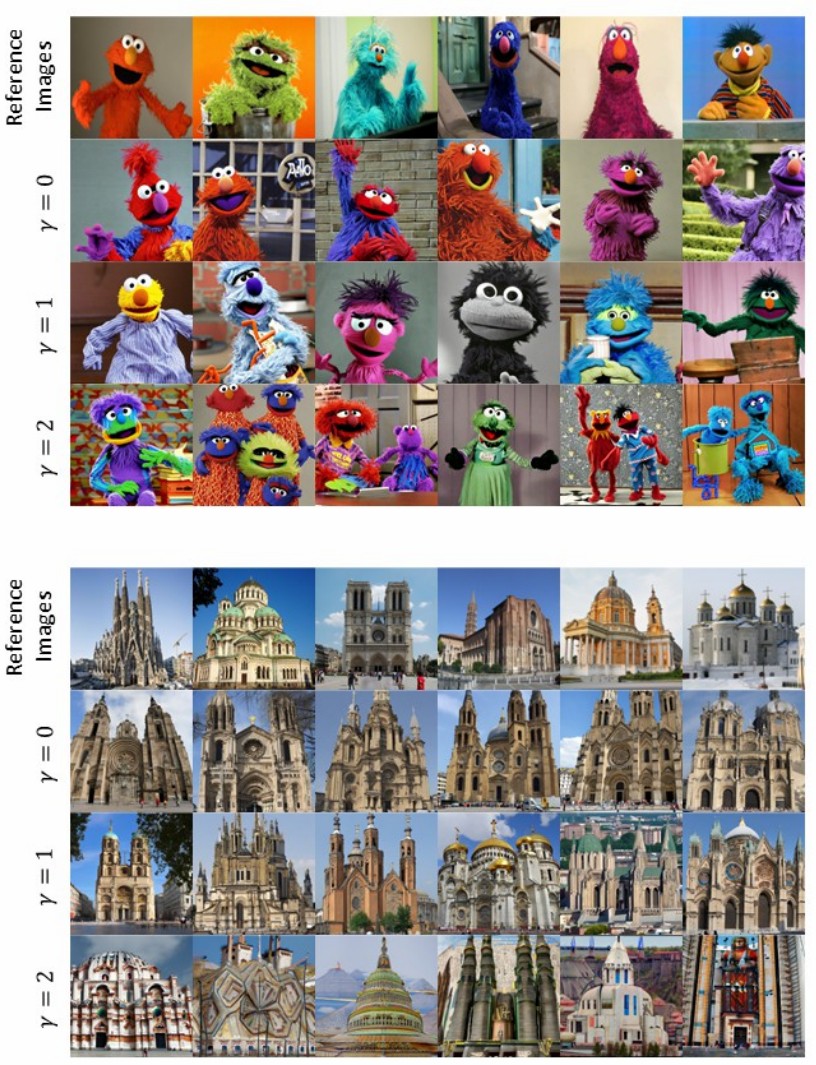

Figure 15: More results on scaling standard deviation of learned prompt distribution.

| Image Set | Method | CLIP-I↑ | DINO↑ | Density↑ | Coverage↑ | FID↓ |
|---|---|---|---|---|---|---|
| French Bulldog | Textual Inversion | 0.82 | 0.49 | 0.10 | 0.12 | 98.76 |
| | DreamBooth | **0.85** | 0.56 | 0.02 | 0.04 | 100.24 |
| | Custom Diffusion | 0.84 | 0.54 | 0.09 | 0.14 | 98.39 |
| | Ours | **0.85** | **0.61** | **0.23** | **0.34** | **88.81** |
| Dogs | Textual Inversion | 0.75 | 0.20 | **1.00** | 0.56 | 194.74 |
| | DreamBooth | 0.76 | 0.20 | 0.68 | 0.44 | 211.34 |
| | Custom Diffusion | **0.77** | **0.21** | **1.00** | **0.64** | 179.51 |
| | Ours | **0.77** | **0.21** | 0.93 | **0.64** | **173.74** |
| Quadrupeds | Textual Inversion | 0.70 | **0.18** | 1.50 | 0.23 | 214.70 |
| | DreamBooth | 0.69 | 0.16 | **2.35** | 0.23 | 228.39 |
| | Custom Diffusion | 0.70 | 0.15 | 2.20 | 0.21 | 217.58 |
| | Ours | **0.71** | **0.18** | 1.89 | **0.24** | **210.93** |

Table 4: Comparison of our method with baselines on feature-based metrics over different granularities of reference images.

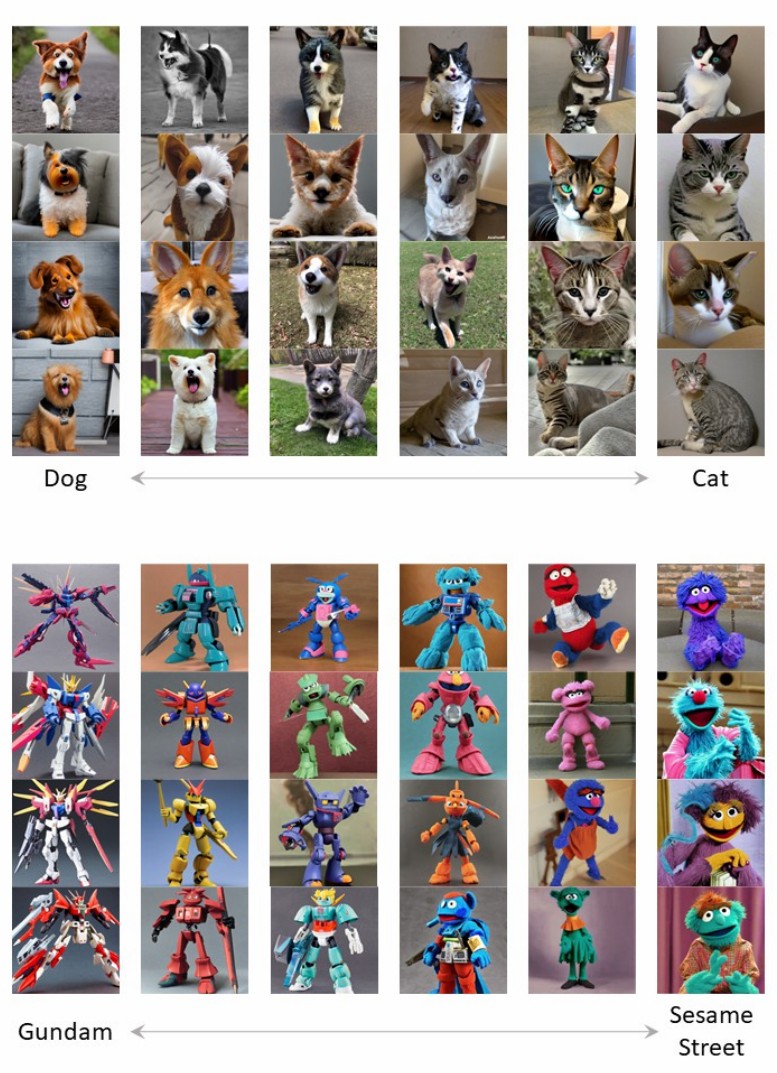

Figure 16: More results on composition of multiple prompt distributions using different weights.

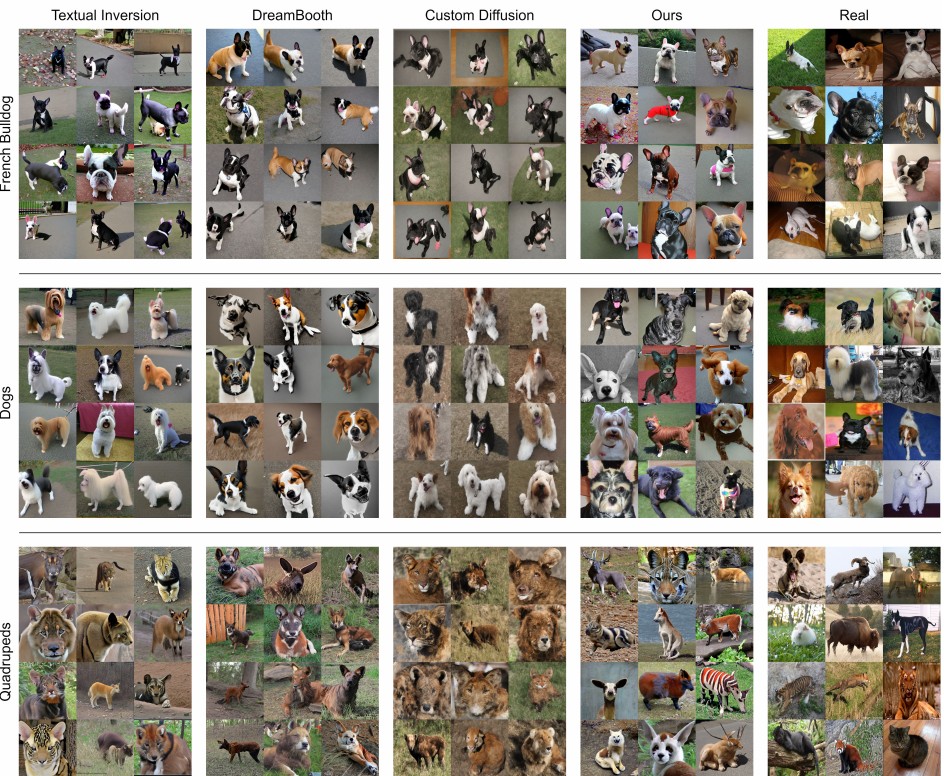

Figure 17: Visualization of reference and generated images using different methods at different class granularity.

fixed learned token to prompt the text-to-3D model, our method samples from the learned prompt distribution, resulting in greater diversity in the generated 3D assets.

Following Corso et al. (2023), we perform a quantitative analysis of in-batch cosine similarities among features of rendered images. Specifically, we compute the average pairwise cosine similarity of DINO and CLIP-I features within each image set. The results shown in Tab. 5 demonstrate that 3D assets generated using our learned prompt distribution exhibit significantly greater diversity compared to those generated using Textual Inversion's learned token, which aligns with visual results. All samples are generated without using random seeds.

| Method | CLIP-I↓ | DINO↓ |
|---|---|---|
| Textual Inversion | 0.63 | 0.31 |
| Ours | **0.53** | **0.14** |

Table 5: Quantitative comparison of 3D generation diversity using averaged pairwise cosine similarity of features of rendered images. Lower similarity means the generated set is more diverse.

## D ABLATION STUDY

**Number of Prompts** We additionally ablate $K$, the number of prompts in personalized generation. We randomly select 4 sets of reference images from our evaluation set and compute the average performance based on automatic quality and diversity metrics introduced in main Section 4.1. In Fig. 8, we show the effect of $K$ in terms of both generation quality and diversity. We observe a positive correlation between the performance (in terms of both quality and diversity) and the number of prompts. More prompts offer more flexibility for our methods to model a better distribution of prompts, thus enabling the model to encapsulate content better (quality) and adapt to various nuances of the training images (diversity).

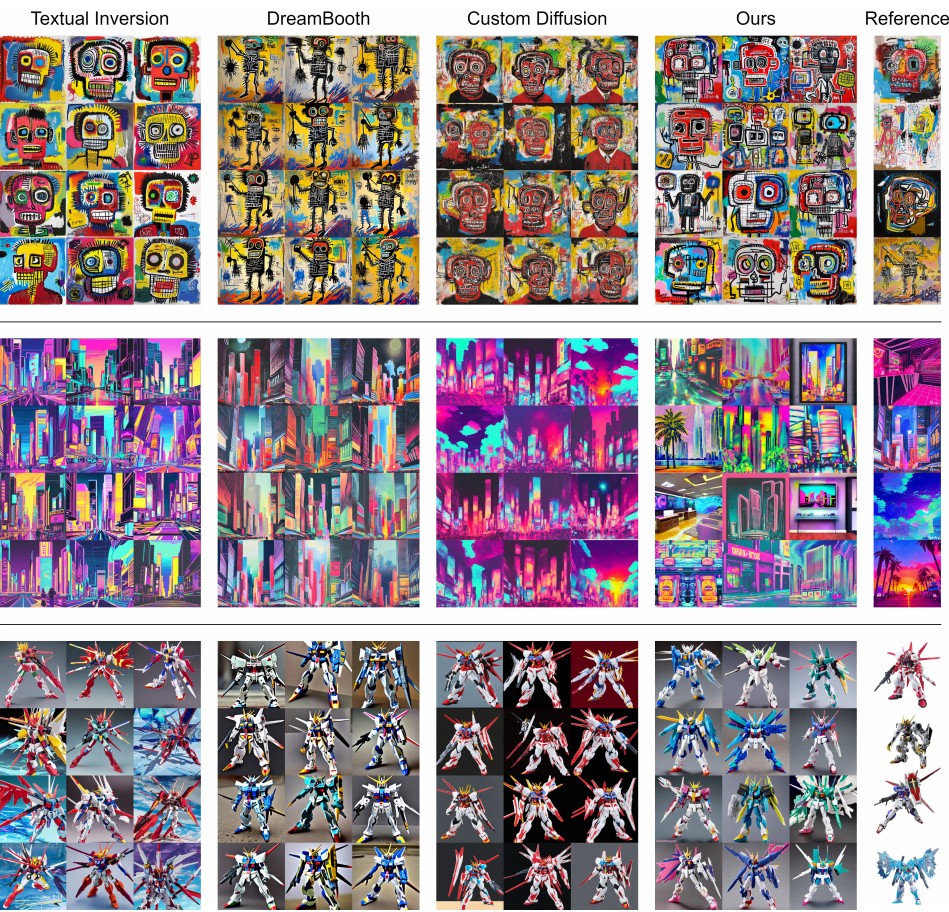

Figure 18: Visual results of training on only 4 reference images.

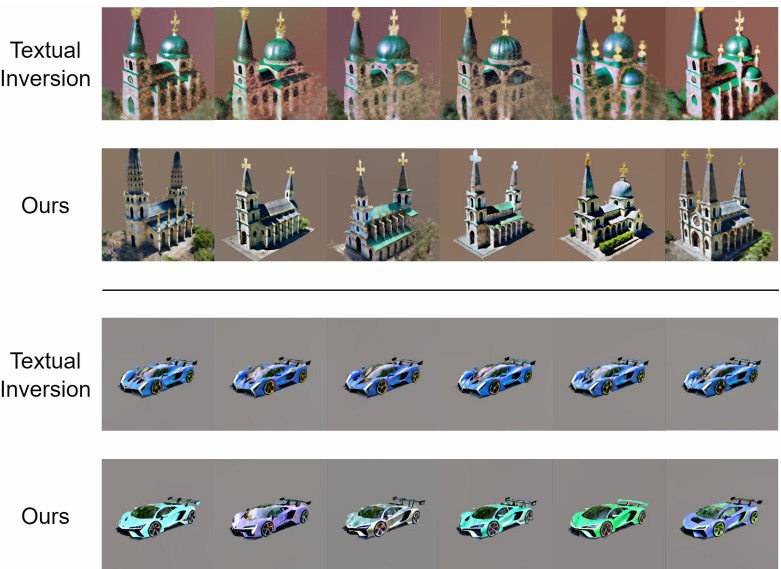

Figure 19: Qualitative comparison of diverse 3D generation.

**Orthogonal Loss** We also test different value of weight of orthogonal loss added during prompt distribution training. As shown in Fig. 9, our choice of $\lambda$ (x-axis) can achieve good balance on quantitative metrics.

| # training images per class | IN | | IN-V2 | | IN-S | | IN-R | | IN-A | |
|---|---|---|---|---|---|---|---|---|---|---|
| | Top1 | Top5 | Top1 | Top5 | Top1 | Top5 | Top1 | Top5 | Top1 | Top5 |
| 10 | 49.8 | 73.8 | 48.5 | 71.4 | 27.7 | 48.8 | 57.1 | 78.1 | 15.3 | 44.4 |
| 100 | 56.7 | 79.8 | **62.0** | 81.5 | **30.4** | **50.9** | **54.7** | **78.4** | **18.8** | **55.0** |
| 500 | **64.3** | **84.0** | 61.7 | **81.6** | 25.2 | 45.8 | 53.0 | 74.8 | 15.7 | 50.4 |
| full(1300) | 63.6 | 82.9 | 60.5 | 80.0 | 18.6 | 36.3 | 43.5 | 67.8 | 11.6 | 45.0 |

Table 6: ImageNet classification accuracy on different real test sets by training a classifier on synthetic ImageNet (IN) generated by DreamDistribution using different number of real training images per class. Our experiments show that using 500 training images per class yields the highest final classification accuracy on real validation set.

**Number of Prompt Token Vectors** We evaluate the impact of varying the number of token vectors $M$ used in each learnable prompt. As shown in Fig. 10, similar to number of prompts, we observe a positive correlation between performance and the number of learnable prompt tokens. However, this effect is less pronounced compared to the impact of changing the number of prompts, and increasing the number of learnable tokens results in longer training times.

# E  SYNTHETIC DATASET

## E.1  MORE ANALYSIS ON SYNTHETIC DATASET

**Number of training images** We experiment with different number of training images. We use randomly selected 10, 100, 500 images per class, as well as all ImageNet training images to train our learnable prompts and generate same size synthetic dataset. From results shown in Tab. 6 and Fig. 20 column 1 & 2, we found that using about 100-500 images per class (7%-38% of the real training set) would be enough to reach high classification accuracy on real validation set, while using more data would not further improve accuracy. For validation sets of ImageNet variants, less training data would obtain higher accuracy due to the domain gap between the real training set and different validation sets.

**Mixing synthetic data with real training data** We also experiment with mixing different sizes of synthetic image data with the real training images. We mix additional 20%, 40%, 60%, 80% and 100% synthetic data with real training data, where 100% means the size of the mixed dataset is twice of the size of the ImageNet training set, and the ratio of the number of real images to the number of synthetic images is approximately 1:1. As shown in Fig. 20 column 3 & 4, adding more synthetic data would improve the accuracy on ImageNet validation set. On the validation sets of different domains, however, adding more synthetic data would not show significant improving trends on accuracy.

## E.2  IMPLEMENTATION DETAIL

For training on ImageNet dataset, we use the same training hyperparameters except for reducing the number of learnable prompts to 10 per class. We train for 5 epochs for training prompt distribution and 300 epochs for training ResNet-50 using generated or mixed dataset. All results are averaged over 3 runs of training using the generated or mixed dataset. For ImageNet-R and ImageNet-A, we only evaluate on the overlapping classes with ImageNet-100.

## E.3  MORE VISUAL RESULT

We show some generated training images using our method and compare them with generated images using baseline methods in Fig. 21, Fig. 22, Fig. 23, Fig. 24, Fig. 25, Fig. 26. Compared to the images generated using baseline methods, non-learning methods could generate images with diversity, however, the appearances are generally far from real images. Learning based personalization baselines can generate images with high fidelity, but with very limited diversity. Our method in contrast can generated images that looks more like real image samples and at the same time with significant diversity.

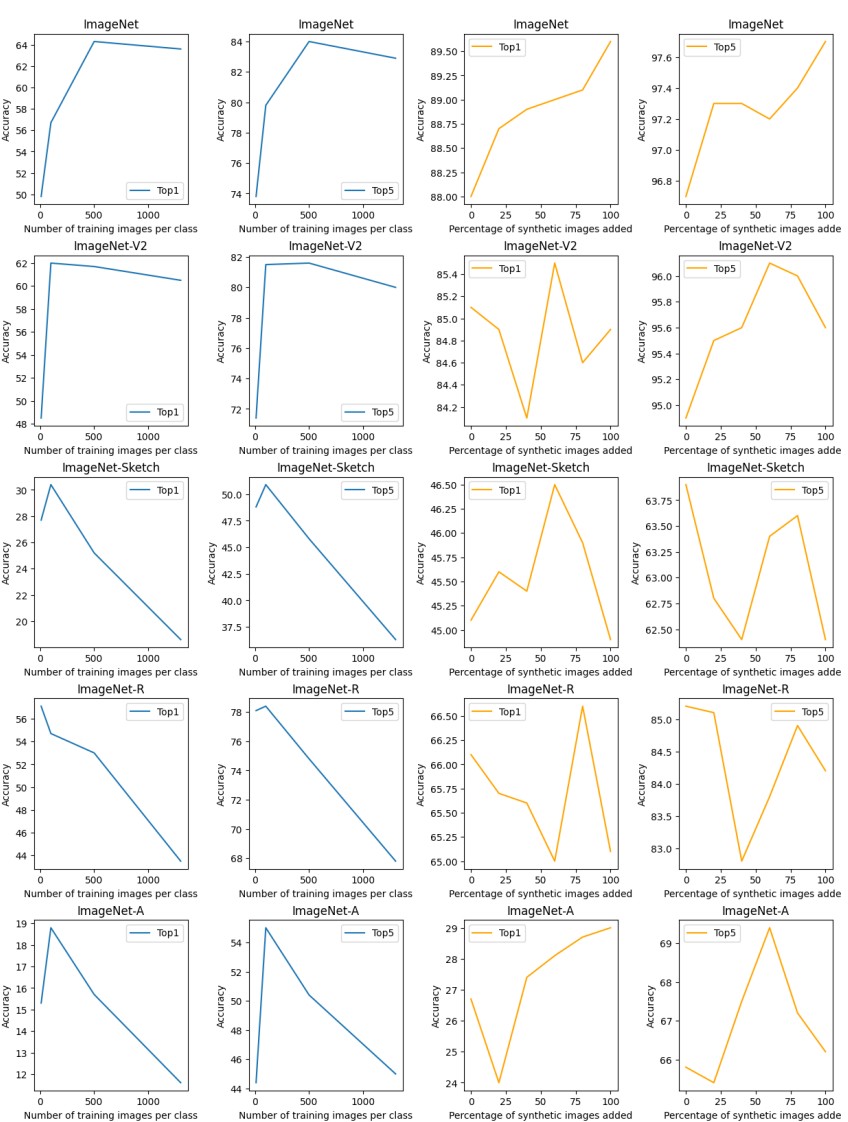

Figure 20: Column 1 & 2: Top-1 and top-5 accuracy on ImageNet validation set versus using different number of training images to train prompt distribution. Column 3 & 4; Top-1 and top-5 accuracy on ImageNet validation set versus percentage of synthetic images added to the real training set.

# F  LIMITATIONS

Despite the ability of our method to generate diverse novel in-distribution images, it does have certain limitations. Specifically, our method may struggle to capture visual features when the number of training images is limited and very diverse. Moreover, the Gaussian distribution assumption could be overly restrictive depending on the training images and the text encoder's latent space. In the future, we hope to find a more robust approach to learning distributions from a few, highly diverse images, with more accurate assumptions and resilient distribution forms.

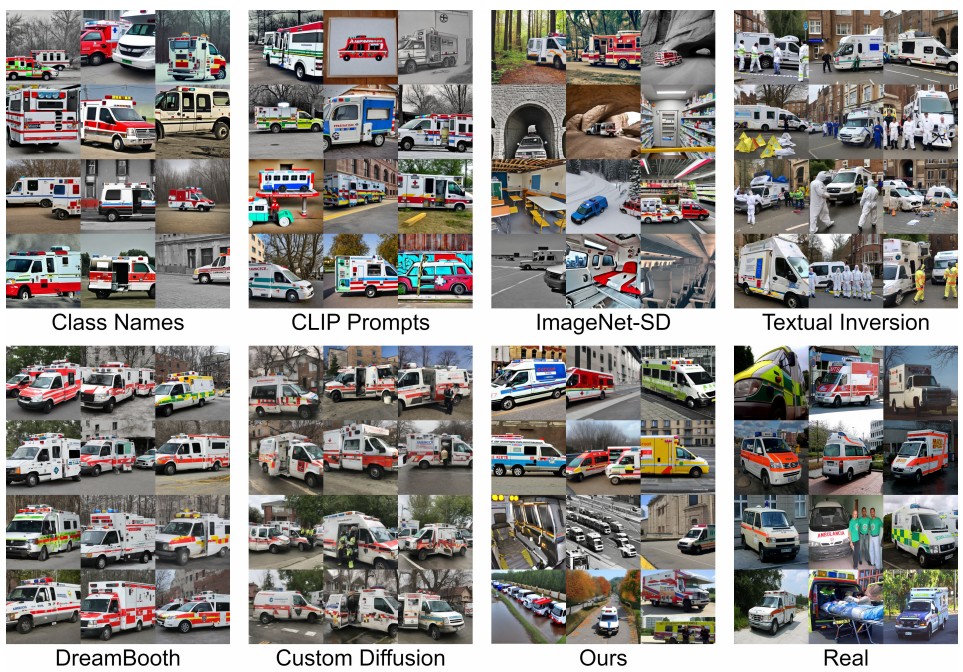

Figure 21: Visualization of generated images on ImageNet ambulance.

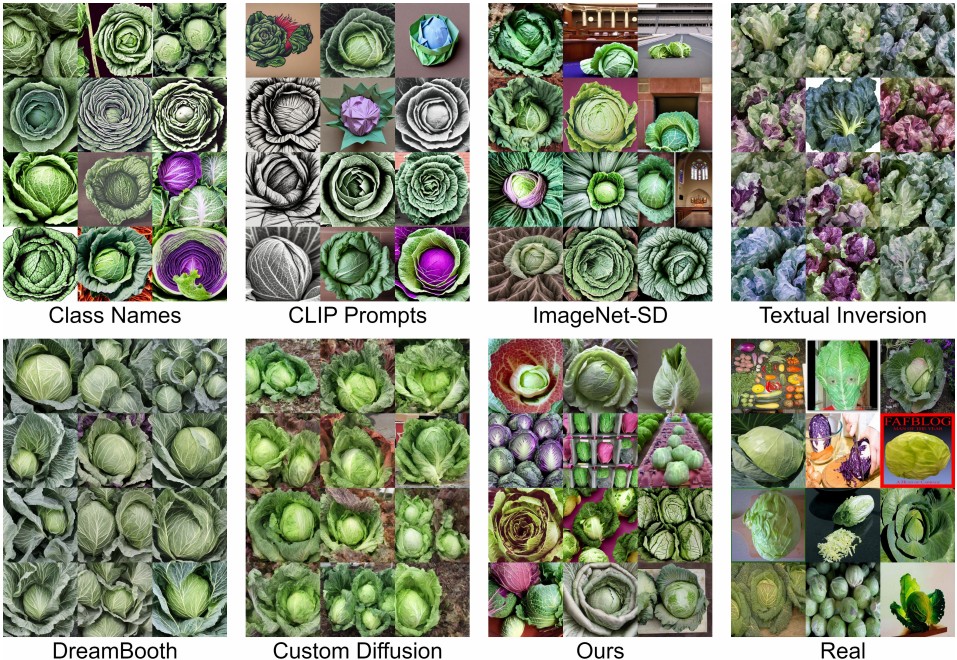

Figure 22: Visualization of generated images on ImageNet cabbage.

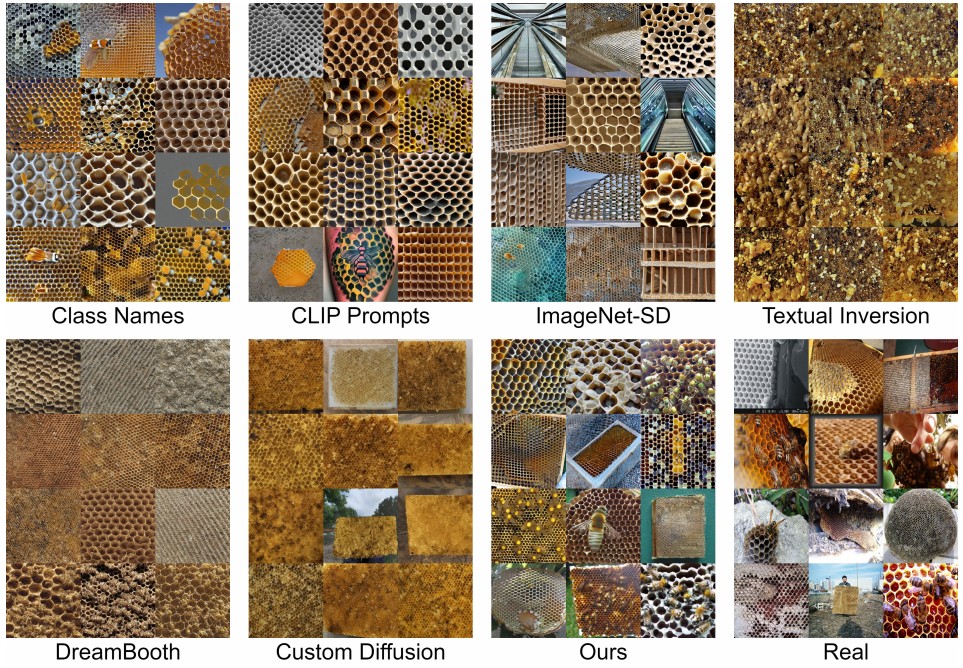

Figure 23: Visualization of generated images on ImageNet honeycomb.

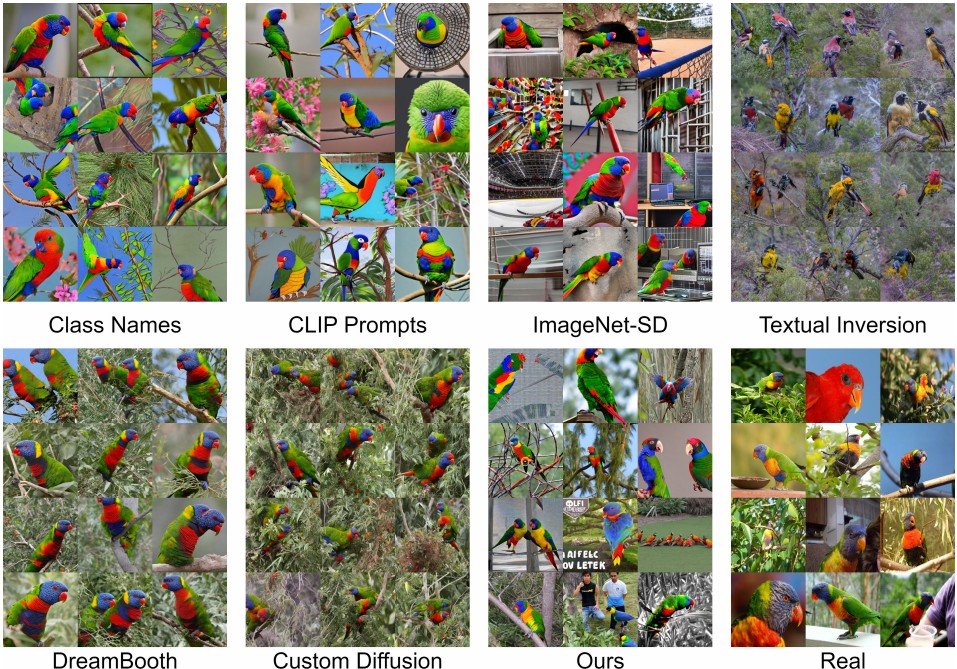

Figure 24: Visualization of generated images on ImageNet lorikeet.

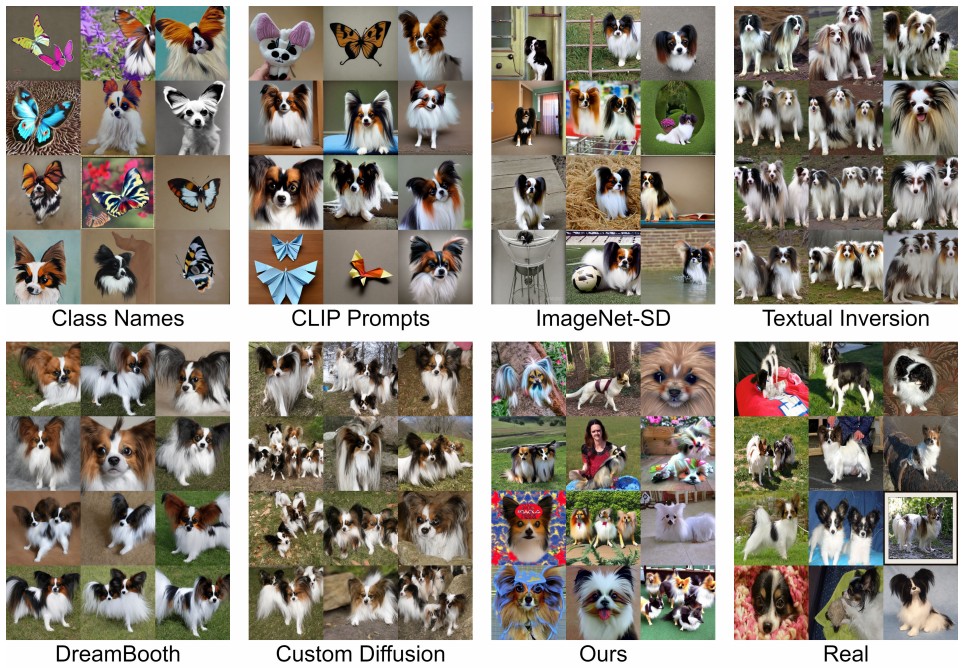

Figure 25: Visualization of generated images on ImageNet papillon.

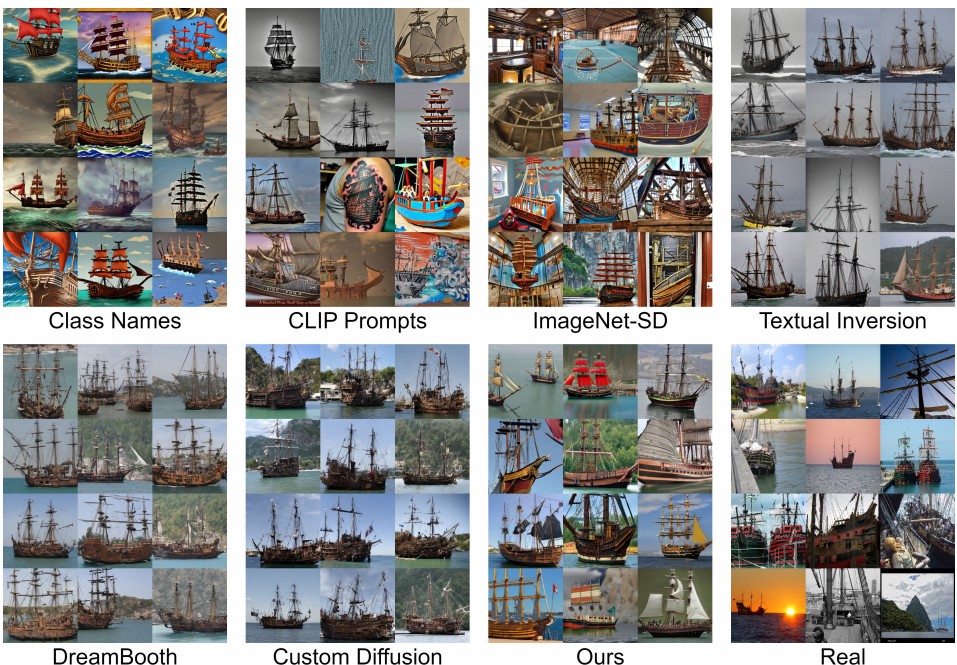

Figure 26: Visualization of generated images on ImageNet pirate ship.

