# OpenReview forum: "DreamDistribution: Learning Prompt Distribution for Diverse In-distribution Generation"
_ICLR.cc/2025/Conference — ICLR 2025 Poster_

### Official Review · Reviewer_TdJX · 2024-10-22

**Soundness:** 3
**Presentation:** 2
**Contribution:** 3
**Rating:** 6
**Confidence:** 4

**Summary:**

The paper proposes a method for personalized image generation. Unlike prior approaches that rely on a fixed text embedding, this paper introduces a probabilistic formulation that samples from a prompt distribution, parameterized by a set of learned soft prompts. They construct a benchmark containing 12 visual concepts (e.g., instances of real objects and illustrations). They also provide a few qualitative examples on text-guided editing, composing prompt distributions, and text-to-3D. Finally, they use their method to generate a synthetic variant of ImageNet, where they demonstrate that a classifier trained on their synthetic images outperforms those produced by prompting or other personalization baselines.

**Strengths:**

- The method is novel and clearly explained; the method introduces a probabilistic formulation that helps with sample diversity. The paper is also careful about the scope of its claims; it studies “different instances in a same category set” not “same-instance personalization” (L357-358).
- The synthetic data generation experiment in Sec 4.5 is a nice metric. Existing personalization methods perform much worse than text-only prompting for producing synthetic data, but the proposed method is able to perform better.

**Weaknesses:**

- The paper could benefit from providing more details about Sec 4.5.
    - For each class, how many images was each personalization method trained on?
    - What is the distribution of noise seeds used to generate the images in Table 1 (e.g., is each synthetic image derived from an independent and completely random sample)?
    - It would be nice if the paper quantitatively verified the claim in L528 (other methods perform worse because they “show limited diversity”) with a diversity metric.
    - It would be nice if the paper provided a qualitative figure similar to Figure 3 for the ImageNet classes.
    - In Table 1, I would be curious to see a synthetic “upper bound” where one uniquely captions each real ImageNet image and generates images from this diverse pool of captions.
    - In Table 1, I would be curious to see a personalization baseline that introduced some randomness, e.g., textual inversion with Gaussian noise added to the learned text embedding.
- The paper could benefit from more clearly stating the types of visual concepts that are in scope for this work.
    - It would be nice if the investigation personalizing to ImageNet classes were extended to class hierarchies. For example, the paper could compare results when considering the class granularity to be “animal” vs. “dog” vs. “Rough Collie” vs. “Lassie”. Then it would be easier to see at what granularity the probabilistic formulation is vs. is not necessary.

**Questions:**

- Below are a few minor comments.
    - L375: missing space; should be “…text-editing **generation. We**…”
    - L413: should be “Text-**guided** Editing”
    -  L50-53 is very tangentially related to the rest of the paper; I think the first paragraph of the introduction could benefit more from the motivation and technical context of the paper.

---

> ### Author Response · Authors · 2024-11-22
>
> Thank you for your time and valuable review!
>
> # Number of Images
> > For each class, how many images was each personalization method trained on?
>
> Thanks for the suggestion, we have updated this information in appendix Figure 11 in the updated version.
>
>
> # Noise Seeds
> > What is the distribution of noise seeds used to generate the images in Table 1 (e.g., is each synthetic image derived from an independent and completely random sample)?
>
> We do not use any random seed in any experiments. Each image is generated independently and completely random. For our method, each image is generated using a random sample from learned prompt distribution.
>
>
> # Diversity Metrics
> > It would be nice if the paper quantitatively verified the claim in L528 (other methods perform worse because they “show limited diversity”) with a diversity metric.
>
> We already show results with diversity metrics in Figure 4(a), and conducing experiments on ImageNet classification is just meant to be show the diversity using synthetic ImageNet as another quantitative proxy other than common feature-based metrics. However, we are happy to add results on ImageNet dataset, also shown in updated Table 1 in our paper.
>
>
> | Method | CLIP-I ↑ | DINO ↑ | Density ↑ | Coverage ↑| FID ↓ |
> | -------- | -------- | -------- | -|-|-|
> | Class Names | 0.75 | 0.43 | 0.71 | 0.41 | 149.87 |
> | CLIP Prompts | 0.71 | 0.37 | 0.63 | 0.40 | 166.42 |
> | ImageNet-SD | 0.72  | 0.38 | 0.54  | 0.36 | 173.58 |
> | Textual Inversion | 0.71 | 0.35 | 0.43 | 0.21 | 214.14 |
> | DreamBooth | 0.77 | 0.50 | 0.89 | 0.51 | 139.49 |
> | Custom Diffusion | 0.75 | 0.47 | 0.84 | 0.42 | 141.53 |
> | Ours | 0.78 | 0.48 | 0.89 | 0.56 | 131.60 |
>
>
>
> # Qualitative Figure for ImageNet
> > It would be nice if the paper provided a qualitative figure similar to Figure 3 for the ImageNet classes.
>
> Thanks for the suggestion, we have added comparison figures in the appendix, please see Figure 20-25 in the updated version.
>
>
> # Synthetic Upper Bound
> > In Table 1, I would be curious to see a synthetic “upper bound” where one uniquely captions each real ImageNet image and generates images from this diverse pool of captions.
>
> The synthetic upper bound is intriguing to consider. However, while maintaining consistency in the image generation model across methods is feasible, the outcomes are likely to be heavily influenced by the choice of image captioning method and its specific outputs. This variability limits its reliability as a stable baseline unless a standardized dataset or fixed prompts for generation, such as ImageNet-SD, is established for reference.
> Again, we agree that this is an interesting experiment, but it needs to be carefully formulated due to the aforementioned reasons. Given the limited time for rebuttal, we may not be able to provide these results at this stage, though we will consider it in our future work if possible.
>
>
> # Baseline with Randomness
> > In Table 1, I would be curious to see a personalization baseline that introduced some randomness, e.g., textual inversion with Gaussian noise added to the learned text embedding.
>
> Please see the table below, where TI+0.5 noise represents the method of adding Gaussian noise with 0.5 variance to the learned Textual Inversion tokens. We also show visualization of Textual Inversion + noise in updated appendix Figure 13.
>
> | **Method** | **CLIP-I** ↑ | **DINO** ↑ | **Density** ↑ | **Coverage** ↑ |  **FID** ↓ |
> |-|-|-|-|-|-|
> | TI+0.5 noise | 0.69            | 0.24       | 0.67          | 0.48           | 368.58 |
> | Ours       | **0.84**       | **0.50**   | **1.59**      | **0.93**       | **215.15** |
>
>
> # Different Granularity
> > It would be nice if the investigation personalizing to ImageNet classes were extended to class hierarchies. For example, the paper could compare results when considering the class granularity to be “animal” vs. “dog” vs. “Rough Collie” vs. “Lassie”. Then it would be easier to see at what granularity the probabilistic formulation is vs. is not necessary.
>
> Thanks for the suggestion, we have added a paragraph in appendix Figure 17 that shows generation results on different granularity of concepts, i.e. French bulldog, different dog species, and different quadruped animals.

---

> > ### Comment · Reviewer_TdJX · 2024-11-22
> >
> > Thank you for taking the time to review my concerns and questions.
> >
> > I especially appreciate the figures you curated in Fig. 20-25 in the Appendix as well as the revision to Table 1 in the main paper; I think this is a nice demonstration that DreamDistribution is useful for producing more diverse images compared to prior work.
> >
> > For the point **Different Granularity**, where I have reviewed the revised Sec. C.7 of the Appendix and Figure 17, I was wondering if the authors could consider the following?
> > - Revise Fig. 17 to also include the strongest baseline, Dreambooth (from Table 1)
> > - Collect a numerical result similar to Table 1, specifically something that looks like
> >
> > | Method             | CLIP-I | DINO   | Density | Coverage | FID    |
> > |--------------------|--------|--------|---------|----------|--------|
> > |**French Bulldog**|
> > | Dreambooth         | - | - | -  | -  | - |
> > | DreamDistribution  | - | - | - | -   | -|
> > |**Dogs**|
> > | Dreambooth         | - | - | -  | -  | - |
> > | DreamDistribution  | - | - | - | -   | -|
> > |**Quadrapeds**|
> > | Dreambooth         | - | - | -  | -  | - |
> > | DreamDistribution  | - | - | - | -   | -|
> >
> > It might be the case that you need to average over more ImageNet classes at each granularity level to get a more convincing result, which is also fine.
> >
> > The main reason why I make this comment is that the primary problem targeted by the work is that it studies “different instances in a same category set” not “same-instance personalization.” **I think a nuanced assessment of the category granularity at which this claim is true, would strengthen the paper.**

---

> > > ### Author Response · Authors · 2024-11-25
> > >
> > > Thank you for your prompt response! We completely agree that assessing granularity is crucial. As also mentioned in our limitations, reference image set that is too diverse may be challenging for our method in capturing an effective distribution.
> > >
> > > We have updated Section C.7 in the appendix, along with Figure 17, which provides visual comparison with baselines, and Table 4, which presents the metric score comparison. We encourage you to review these and please let us know if you have any further concerns or questions.

---

> > > > ### Comment · Reviewer_TdJX · 2024-11-26
> > > >
> > > > Thank you for taking the time to respond to my questions and for adding the requested quantitative result to Sec. C.7 to the Appendix. I have raised my score to 6 accordingly.

---

> > > > > ### Author Response · Authors · 2024-11-27
> > > > >
> > > > > Thank you so much for your constructive review!

---

### Official Review · Reviewer_BPjf · 2024-11-02

**Soundness:** 3
**Presentation:** 3
**Contribution:** 3
**Rating:** 6
**Confidence:** 4

**Summary:**

This paper proposes distribution learning in the prompt space to increase the diversity for personalized generation problems.
The effectiveness of the proposed method has been proved on text-to-image, text-to-3D and synthetic dataset generation.

**Strengths:**

1. The writing of this paper is good and easy to follow.
2. The proposed soultion for increasing the generation diversity is straightforward and effective.
3. With application to different scenarios (text to image, text to 3D, and synthetic dataset generation), the authors provide different perspectives to demonstrate the effectiveness of the method.

**Weaknesses:**

1. The methods compared in this manuscript are no longer the state-of-the-art for personalized generation.  It would be better if the authors can compare with more recent and widely used methods, eg, IP-Adapter.
2. The main contribution of this paper happens in the prompt space. However, I didn't see too much discussion and experiments that focus on the prompt space. For example, how does the orthogonal loss affect the generation quality and diversity? How different the K learnable prompts will be if the prompt distribution learning is not used?
3. I didn't see either qualitative or quantitative comparisons with our personalized text-to-3D methods, although the authors claim the diversity is better.

**Questions:**

Please refer to the weakness part.

---

> ### Author Response · Authors · 2024-11-22
>
> Thank you for your time and valuable review!
>
> # Other Baselines
> > The methods compared in this manuscript are no longer the state-of-the-art for personalized generation. It would be better if the authors can compare with more recent and widely used methods, eg, IP-Adapter.
>
> Thank you for the suggestion. Below, we present the quantitative comparison results with IP-Adapter. If accepted, we will try our best to include IP-Adapter comparisons for all other experiments and figures in the camera-ready version.
>
> | **Method** | **CLIP-I** ↑ | **CLIP-T** ↑ | **DINO** ↑ | **Density** ↑ | **Coverage** ↑ | **FID** ↓ |
> |-|-|-|-|-|-|-|
> | IP-Adapter | 0.80         | 0.23         | 0.46       | 0.99          | 0.91           |  216.99  |
> | Ours       | **0.84**     | **0.29**     | **0.50**   | **1.59**      | **0.93**       | **215.15** |
>
>
> # Prompt Space Analysis
> > The main contribution of this paper happens in the prompt space. However, I didn't see too much discussion and experiments that focus on the prompt space. For example, how does the orthogonal loss affect the generation quality and diversity? How different the K learnable prompts will be if the prompt distribution learning is not used?
>
> - **Ablation of orthogonal loss**: We conducted ablation studies by varying orthogonal loss weights. Since it is challenging to qualitatively assess subtle image set similarity and diversity across different weights, we provide quantitative results in Appendix Figure 9, including the case with zero orthogonal loss.
> - **Analyzing final prompt variance**: Additionally, we analyzed the average final prompt variance of 32 learnable prompts as a function of the orthogonal loss weight, highlighting its significant impact on the resulting distribution. While the variance shows an increasing trend with higher orthogonal loss, it is also influenced by factors such as the diversity of reference images, learning rate, and random initialization of parameters.
>
> | Orthogonal Loss Weight | Final Prompt Variance |
> | -------- | -------- |
> | 0     | 0.0017     |
> | 1e-4 | 0.0020 |
> | 1e-3 | 0.0042 |
> | 1e-2 | 0.0062 |
> | 1e-1 | 0.0073 |
> | 1 | 0.0093 |
>
>
>
> # 3D Comparisons
> > I didn't see either qualitative or quantitative comparisons with our personalized text-to-3D methods, although the authors claim the diversity is better.
>
> Thanks for the suggestion, since evaluating set-level metrics need to generate more than tens or hundreds of instances, while generating one 3D model needs more than one hour, compared with just less than a few seconds to generate one image, we primarily focus on evaluation and visualization on images instead of 3D.
> Following your suggestion, we tried our best to generate an evaluation set with 6 generations for each learned token/distribution using our method and Textual Inversion applied to MVDream, and we show the comparison in appendix Figure 19. Quantitatively, we calculate cosine similarity of CLIP-I and DINO features (lower means more diverse) of 3D renderings of same class to show the diversity comparison of generated 3D assets. The table is also added to the appendix.
>
>
> | Method | CLIP-I↓ | DINO↓ |
> | - | -------- | -------- |
> |Textual Inversion| 0.63 |  0.31  |
> |Ours| **0.53** | **0.14** |
>
> Edit: Updated figure number

---

> ### Author Response · Authors · 2024-12-02
> **Happy to answer any further questions**
>
> Dear Reviewer BPjf,
>
> Thank you for taking the time to review our submission. We have provided responses to your suggestions and questions and would be happy to address any further concerns or questions. We are looking forward to your feedback.
>
> Best,
>
> Authors of submission 503

---

### Official Review · Reviewer_3gbS · 2024-11-03

**Soundness:** 3
**Presentation:** 3
**Contribution:** 2
**Rating:** 6
**Confidence:** 4

**Summary:**

This paper presents DreamDistribution, a solution for personalized text-to-image diffusion-based image generation. The key goal is to model the textual distribution of a training set, and guide the diffusion models to sample images that are adhere to the semantics of the reference images. Some learnable soft textual prompts and the reparameterization trick are introduced, enabling the generation of novel images by sampling prompts from the learned distribution. The authors also show the adaptability of the proposed DreamDistribution by applying it to various downstream tasks. Quantitative and qualitative analysis demonstrates the superiority of DreamDistribution against competing tuning based methods.

**Strengths:**

1. The writing of the paper is clear and easy to follow.
2. Extensive visual results are shown for intuitive comprehension of the proposed method.
3. The intuition presented in the paper is simple yet quite straightforward, supported by the comprehensive comparative experiments.
4. The proposed method only requires maintaining and updating a few learnable text prompts, which greatly preserves the image prior of the diffusion model.

**Weaknesses:**

1. The model relies heavily on the text-image alignment of diffusion, and since the diffusion model itself has not been tuned, I am curious about whether DreamDistribution can generate the new concepts if the diffusion model has not seen the reference images before.
2. I did not observe a significant visual improvement compared to textual inversion. And maybe the performance gap will become even smaller when stronger base models are utilized.
3. The novelty of proposed method is quite limited. Learnable tokens are very common in NLP and visual-language tasks, and the orthogonal loss is borrowed from some dictionary learning based methods, the reparameterization trick is borrowed from VAE training. These narrow the technical contribution of the proposed DreamDistrition.
4. DreamDistribution requires large amount of reference domain images to converge. I wonder how it performs when training samples are limited to 1~10, and how can it beats textual inversion in such scenarios.
5. The method is based on the assumption that the text distribution corresponding to a concept follows a Gaussian distribution. Is this a reasonable assumption? Also, does the number of learnable prompts required for different concepts need to be the same?

**Questions:**

Please refer to the Weakness.

---

> ### Author Response · Authors · 2024-11-22
>
> Thank you for your time and valuable reviews!
>
> # Generation of New Concepts
> > The model relies heavily on the text-image alignment of diffusion, and since the diffusion model itself has not been tuned, I am curious about whether DreamDistribution can generate the new concepts if the diffusion model has not seen the reference images before.
>
> This is an interesting question. If we understand it correctly, it also applies to Textual Inversion, where only the prompt is tuned, and the diffusion model itself has likely never seen the concept that the user wants to personalize in the reference images.
>
> 1. Our method is capable of learning new concepts from reference images. For example, as shown in the Gundam example in Figure 3 short caption baseline method, where diffusion model is prompt with text only. The diffusion model generates a concept of Gundam that differs from the reference images, indicating that the Gundam concept in the reference images is new to the diffusion model. However, after training on the reference images, our method can generate Gundam images that closely resemble this new Gundam concept shown in the reference images. This demonstrates how our method learns a new concept of Gundam from reference images that is distinct from what the diffusion model originally interprets Gundams to look like.
> 2. Intuitively, text-to-image diffusion model, having been trained on a wide variety of images and their corresponding captions, has learned a rich, semantically structured distribution of what is considered a meaningful image by humans. Both Textual Inversion and our method work by learning a soft prompt or a distribution of soft prompts in a continuous space that positions the new concept within this distribution, effectively mapping it onto existing features. This learning process does not introduce new information into the diffusion model but adapts the prompt to leverage the model’s existing knowledge to approximate the new concept. As a result, the learned soft prompt distribution enables the diffusion model to generate images that are similar to the reference set without requiring the model to have been directly trained on those specific images.
> 3. Moreover, learning the prompt instead of tuning the diffusion model’s weights helps preserve the generalizability of the original diffusion model. In contrast, tuning the diffusion model is prone to overfitting to the reference images, which limits its ability to generate outputs with different text prompt guidance, as stated in the DreamBooth paper.
>
>
>
>
> # Visual Improvement
> > I did not observe a significant visual improvement compared to textual inversion. And maybe the performance gap will become even smaller when stronger base models are utilized.
>
> Could the reviewer clarify the term "visual improvement"?
> If "visual improvement" refers to the quality of generation results based solely on human perception, independent of reference images, then our method does not directly control that aspect, as it primarily focuses on training input prompts for the diffusion model. However, our method is applicable to various base models, as long as they have a text encoder capable of passing gradients back to the text embedding space.
> If "visual" refers to diversity and similarity compared to the reference images, Figure 4 in our paper, as well as the synthetic dataset experiment in Section 4.5, provide quantitative results demonstrating the effectiveness of our method.
> We have also added more visualizations in Appendix Figure 20-25, which highlight that images generated by other personalization baselines exhibit very limited diversity.

---

> ### Author Response · Authors · 2024-11-22
>
> # Contribution
> > The novelty of proposed method is quite limited. Learnable tokens are very common in NLP and visual-language tasks, and the orthogonal loss is borrowed from some dictionary learning based methods, the reparameterization trick is borrowed from VAE training. These narrow the technical contribution of the proposed DreamDistrition.
>
> To the best of our knowledge, we are the first to apply prompt distribution learning to visual generative tasks, enabling set-level personalized generation with diversity. Traditional baseline methods like Textual Inversion and DreamBooth are limited to instance-level personalization with restricted diversity. In contrast, our method not only preserves the instance-level personalization capability but also effectively extends to personalization at the set or category level.
>
> To tackle the novel set-level diverse and personalized generation task, the implementation and experiments are also non-trivial:
> 1. Learnable text tokens for image generation were first introduced in Textual Inversion, inspired by the concept of GAN inversion. However, Textual Inversion is not well-suited for scenarios where user-provided images are diverse and do not represent a single image, instance, or concept.
> 2. To address this new problem, we introduce prompt distribution learning to capture a diverse representation comprising multiple learnable prompts rather than a fixed one. While our approach and dictionary-based learning share some similarities, such as the use of orthogonal loss to ensure distinctiveness among learned elements, they fundamentally differ in purpose and application. Dictionary-based learning focuses on efficient data compression or reconstruction by learning a set of bases that can best reconstruct data through combinations. In contrast, our method models a semantic distribution of image representations and relies on random sampling for generation. Our application of orthogonal loss specifically promotes semantic separation and diversity within the language latent space, enabling varied prompt representations to be sampled from the learned distribution, unlike the traditional focus of dictionary learning on effective signal compression and reconstruction.
> 3. The reparameterization trick is essential in our pipeline to enable gradient backpropagation in distribution learning methods. However, simply applying this trick is insufficient in our case. As detailed in our method section 3.3 and pseudocode Algorithm 1 in appendix, our objective function lacks a closed-form solution, necessitating the use of a Monte Carlo approach to approximate the expected value for optimization.
> 4. Our approach represents a significant and impactful application in the realm of prompt distribution learning for diverse and personalized image generation. Extensive qualitative and quantitative experiments highlight the effectiveness of our method in achieving superior diversity and quality in this challenging task.
>
> Furthermore, reviewer TdJX describes our method as "novel and clearly explained", and reviewer BPjf calls it "straightforward and effective". Based on these assessments, we humbly believe our method is novel, simple, and effective.
>
>
>
> # Less Sample Performance
> > DreamDistribution requires large amount of reference domain images to converge. I wonder how it performs when training samples are limited to 1~10, and how can it beats textual inversion in such scenarios.
>
> The diversity of generated outputs in our method is indeed influenced by the number and variety of reference images, as outlined in our limitations section. With fewer or less diverse reference images, the resulting outputs will similarly lack diversity if other factors remain unchanged.
> In the extreme case of a single reference image, both our method and baseline methods will focus on replicating that image closely. However, our method includes a hyperparameter-controlled orthogonal loss, introducing additional randomness into the learned prompt, which contributes to the diversity of the generated results.
> In our experiments, each reference set typically contains 10–20 diverse images. Here, we present results with reduced reference sets of 5 images, keeping all other settings unchanged. It is important to note, however, that fewer reference images lead to less stable and reliable results.
>
> |**Method**| **CLIP-I** ↑ | **DINO** ↑ | **Density** ↑ | **Coverage** ↑ | **FID** ↓ |
> |-|-|-|-|-|-|
> |Textual Inversion|  0.84  |**0.50**|  1.29  |  0.82        |  313.47  |
> |Ours             |**0.85**|  0.47  |**1.48**|**0.88**      |**291.62**|

---

> ### Author Response · Authors · 2024-11-22
>
> # Gaussian Assumption
> > The method is based on the assumption that the text distribution corresponding to a concept follows a Gaussian distribution. Is this a reasonable assumption?
>
> Gaussian assumption is commonly used and proved in prior works: [1] shows that the output embeddings of CLIP text encoder for an ImageNet category are close and can be modeled by Gaussian. [2] claims text embeddings describing same concept are clustered and follow Gaussian for both language (BERT, GPT) and vision-language models (CLIP). [3] shows diverse learned prompts within one class are well-clustered showing it can be modeled by Gaussian. A term to induce soft prompts is not needed. As discussed in the limitations, Gaussian assumption may be not optimal when reference images that are highly diverse, e.g., a mixture of multiple categorical images, for which a more sophisticated distribution might be needed. However, a Gaussian is sufficient in our work to learn common concept of an image set for personalized generation. Our qualitative and quantitative results also prove the effectiveness.
>
> [1] Lu, Yuning, et al. "Prompt distribution learning." Proceedings of the IEEE/CVF Conference on Computer Vision and Pattern Recognition. 2022.
>
> [2] Ma, Wenxuan, et al. "Borrowing Knowledge From Pre-trained Language Model." Proceedings of the IEEE/CVF international conference on computer vision. 2023.
>
> [3] Chen, Guangyi, et al. "PLOT: Prompt Learning with Optimal Transport for Vision-Language Models." The Eleventh International Conference on Learning Representations. 2023.
>
>
>
> # Number of Learnable Prompts
> > Also, does the number of learnable prompts required for different concepts need to be the same?
>
> The number of learnable prompts is a hyperparameter. From our ablation results shown in appendix section D, we observed that increasing the number of learnable prompts generally improves performance. However, this comes at the cost of higher training time and memory usage. For our experiments, we set $K=10$ for ImageNet experiments and $K=32$ for all other experiments.

---

> ### Comment · Reviewer_3gbS · 2024-11-25
>
> Thank you for answering my questions. Can you please show me some visual results when given samples less than 5? I'm still wondering whether DreamDistribution is a user-friendly method when utilized as a personalization method. For most generative customization scenarios, we can't ask the users to provide dozens of images (50-100) to describe the concepts they want to generate.
> Also, as the base image generation model becoming stronger, what is the superiority of DreamDistribution against some instruction based methods? For example, I can leverage a strong vision-language model to describe reference images with image captions, and utilize the captions to generate new images with some SOTA DiT based models (e.g., flux, which supports more complex text descriptions as input), this seems much more users friendly and easier than DreamDistribution.

---

> > ### Author Response · Authors · 2024-11-27
> >
> > Thank you so much for the response! Please see our reply to you questions below:
> >
> > # Less samples
> > > Can you please show me some visual results when given samples less than 5? I'm still wondering whether DreamDistribution is a user-friendly method when utilized as a personalization method. For most generative customization scenarios, we can't ask the users to provide dozens of images (50-100) to describe the concepts they want to generate.
> >
> > Our experiments demonstrate that DreamDistribution performs effectively with around 10-20 images, which we believe strikes a good balance between capturing diversity and maintaining usability. For comprehensiveness, we also experiment with reference sets with only 4 images for each class, and we have updated our appendix C.8, along with visual results in Figure 18 as requested, which shows that our method can still generate results with more diversity and at the same time within the distribution of reference images.
> >
> > # Compare with instruction base method
> > > Also, as the base image generation model becoming stronger, what is the superiority of DreamDistribution against some instruction based methods? For example, I can leverage a strong vision-language model to describe reference images with image captions, and utilize the captions to generate new images with some SOTA DiT based models (e.g., flux, which supports more complex text descriptions as input), this seems much more users friendly and easier than DreamDistribution.
> >
> > 1. **Information loss due to language as medium** Even with advanced captioning methods, concepts in images cannot be perfectly reconstructed due to inherent information loss when converting images to text and back to images. Many attributes, such as camera view, lighting, texture, and spatial relationships, are difficult to describe precisely even with highly detailed captions, making them challenging for text-to-image models to interpret. Here we show a few examples in this anonymous [link](https://ibb.co/zhc1B5N), where we use GPT-4V to generate a detailed caption from the reference image and FLUX for text-to-image generation. In the first example, text-to-image generation fails to accurately reproduce the style, texture, and relative positioning of objects in the painting. In the second example, the caption does not capture the specific pattern of the sauce dots or the precise arrangement of food items. The third example demonstrates how "fuzzy" textures are interpreted differently in generated images. Lastly, the fourth example highlights the difficulty of capturing implicit details such as camera view and lighting in captions. The text-to-image model also generates an incorrect number of domes, even though the exact number is specified in the caption. In contrast, our learning-based method can easily learn the concept by training directly on the single reference image, avoiding the limitations of language as an intermediary. The challenges of caption-based approaches highlights the value of learning-based methods, which operate in a continuous latent space instead of a discrete text token space, effectively mitigating issues of information loss and preserving finer details.
> > 2. **Enable new task** Again we want to emphasize that our method is highly versatile and can be applied to various base generation models. Regardless of advancements in image generation models, the strength of our approach lies in its ability to enable a new set-level personalization task. It surpasses instance-level personalization baselines that focus on reconstructing a single concept and is distinct from instruction-based methods, which we view as addressing a complementary, orthogonal use case.
> > 4. **Orthogonal use case** We believe that the proposed use of caption and instruction approach for generation addresses a use case that is orthogonal to our method. Our approach is best suited for scenarios where users have a few reference images but do not have a specific modification in mind or find it hard to summarize visual attributes using natural language, relying instead on automatically capturing common visual attributes and applying variations to generate creative and plausible outputs. In such cases, our method provides a more effective solution compared to experimenting with captioning methods and various instructions. However, when users have a clear and precise modification they wish to make based on a single image, our method may not be ideal, as sampling from a prompt distribution introduces randomness that might not consistently yield the exact desired result.

---

> > > ### Comment · Reviewer_3gbS · 2024-11-27
> > >
> > > Thank you for your response. Most of my concerns are addressed. I decide to increase my final score. Yet I strongly recommend authors to do further experiments by leveraging stronger base models, because most semantic issues of previous T2I models may not be encountered when using newest models (e.g. DiT based generation models).

---

> > > > ### Author Response · Authors · 2024-12-02
> > > >
> > > > Thank you so much for the update! We appreciate your suggestion and will certainly explore various base models in future work.

---

### Official Review · Reviewer_RW1y · 2024-11-04

**Soundness:** 2
**Presentation:** 3
**Contribution:** 2
**Rating:** 5
**Confidence:** 3

**Summary:**

This paper proposes a method to learn personalized prompts that not only capture the common concept embedded in the given reference images but also reflect variations between them. Specifically, instead of learning just one soft prompt, multiple soft prompts are learned, and their mean and deviation are used to generate diverse input prompts. To ensure that each prompt learns significantly different values, an additional objective function that encourages orthogonality between the prompts was introduced during training. The authors demonstrate that this method successfully learns prompts that reflect diverse characteristics of given reference compared to previous works and suggest that it can be extended to 3D generation models.

**Strengths:**

- The explanation of the target problen, the method proposed to solve it, and the experimental setup are clear.
- The effectiveness of the proposed method is convincingly validated through various experimental results.

**Weaknesses:**

- The baseline methods do not share the target problem of this work. Models like DreamBooth or Custom Diffusion focus on learning only the common concept by eliminating variation, so they may not be fair comparison targets.
- In Figure 5-(a), when only the mean value is used, it seems that this model also fails to generate variable images. This suggests that simply learning one soft prompt and randomly perturbing it might suffice. However, without ablation studies, it is hard to be sure whether the proposed method itself is truly meaningful.
- In task like this, human evaluation metrics are particularly important. However, the result in Figure 4-(b) is based on evaluations from only 10 participants, which undermines reliability.

**Questions:**

- In models like Custom Diffusion, if various text prompts containing the learned special token are used, isn't it possible to generate diverse images?
- What are the results when only one learned soft prompt is used? Can we confirm what characteristics each soft prompt has learned?
- How the results change according the value of M?

---

> ### Author Response · Authors · 2024-11-22
>
> Thank you for the valuable review!
>
> # Target Baselines
> > The baseline methods do not share the target problem of this work. Models like DreamBooth or Custom Diffusion focus on learning only the common concept by eliminating variation, so they may not be fair comparison targets.
>
> Our method addresses a more generalized task than models like DreamBooth, introducing the novel ability to generate diverse instances while maintaining learned common properties. To the best of our knowledge, this is the first work to tackle the challenging task of diverse personalization generation. As we have not come across baselines that directly address this task, we adapted the most relevant related works for comparison. Specifically, we include three learning-based methods: Textual Inversion, DreamBooth, and Custom Diffusion, which are popular, important, and closely related to our setting, as well as two non-learning-based baselines: short captions and long captions. Comparative results are presented in Figure 3 and Figure 4. We believe these are good baselines that closely resemble the task we are trying to solve. We tried our best to optimize these methods for our task. If the reviewer is aware of other methods suited to the same task, we would be happy to include additional comparison experiments.
>
>
>
>
> # Variable Image Generation
> > In Figure 5-(a), when only the mean value is used, it seems that this model also fails to generate variable images. This suggests that simply learning one soft prompt and randomly perturbing it might suffice. However, without ablation studies, it is hard to be sure whether the proposed method itself is truly meaningful.
>
> In Figure 5a, we demonstrate that increasing the variance of the learned prompt results in greater diversity in the generated images. However, this diversity arises from randomness and is not directly related to the reference image sets. An example is provided in Figure 13 in the revised appendix, showing that our method effectively captures the shared attributes of the reference images, such as object shape in the Bearbrick example and object appearance in the water bottle example, while varying only the differences, like texture patterns for Bearbrick and backgrounds for water bottles. In contrast, using a single soft prompt with random perturbations struggles to preserve the common attributes of the reference images, such as maintaining a consistent shape in the Bearbrick example. Please refer to the updated Figure 13 in the appendix for a visualization comparison. Furthermore, in our response to reviewer TdJX titled "Baseline with Randomness", we did a quick classification experiment on 10 classes, and the result shows that our methods still perform better than simply adding some noise to one soft prompt.

---

> ### Author Response · Authors · 2024-11-22
>
> # Human Evaluation
> > In task like this, human evaluation metrics are particularly important. However, the result in Figure 4-(b) is based on evaluations from only 10 participants, which undermines reliability.
>
> We appreciate the reviewer's emphasis on human evaluation metrics.
> 1. **Details on Evaluation Trials**: We define a trial as one participant reviewing 4 image sets generated by our method and 3 baseline methods, and selecting their preferred method as a data sample for the final percentage calculation. Each participant contributed 12 trials by reviewing 12 different classes, resulting in a total of 120 trials across all participants. Since each image set consists of 40 generated images, each trial involved comparing $4×40=160$ images, totaling $12×4×40=1920$ images reviewed in total by one participant. The result from our human evaluation aligns with our quantitative metrics result.
> 2. **Additional Evaluation**: To further enhance reliability, we have increased the participant pool, adding 10 more participants with 120 additional trials, resulting in a total of 240 trials, and we do not observe a significant change in the result, where previously 62.0% of 120 trials preferred ours, and now 63.3% of 240 trials prefer ours. We have updated the result in Figure 4b.
> 3. **Challenges of Human Evaluation for Set-Level Metrics**: Human evaluation for set-level metrics like diversity is inherently challenging. Evaluating concepts such as similarity and diversity across sets requires participants to assess tens or hundreds of images for one trial, making it time consuming and difficult to find volunteers. Many prior works addressing generation diversity omit user preference studies altogether [1, 2, 3, 4, 5]. Although challenging, we tried our best to find volunteers and collect 240 data for our human preference study.
> 4. **Additional Visualizations**: Furthermore, we have included more visualizations using ImageNet images in the appendix Figure 21-26 to provide a clearer sense of our method’s capabilities.
>
>
> [1] Corso, Gabriele, et al. "Particle guidance: non-iid diverse sampling with diffusion models." ICLR 2024.
>
> [2] Sadat, Seyedmorteza, et al. "CADS: Unleashing the diversity of diffusion models through condition-annealed sampling." ICLR 2024.
>
> [3] Zichen Miao, et al. "Training Diffusion Models Towards Diverse Image Generation with Reinforcement Learning." CVPR 2024.
>
> [4] Lu, Yuanxun, et al. "Direct2. 5: Diverse text-to-3d generation via multi-view 2.5 d diffusion." CVPR 2024.
>
> [5] Tran, Uy Dieu, et al. "Diverse Text-to-3D Synthesis with Augmented Text Embedding." ECCV 2024.
>
> Edit: Updated figure numbers, clarified human evaluation details

---

> ### Author Response · Authors · 2024-11-22
>
> # Generate with Diverse Prompts
> > In models like Custom Diffusion, if various text prompts containing the learned special token are used, isn't it possible to generate diverse images?
>
> Using such method is not a capable way to generate large amount of diverse images that resembles the distribution of a set of reference images.
> 1. Models like Custom Diffusion learn a fixed representation tied to a specific instance. Even with diverse reference images, the learned special token converges to a single appearance (such as same red and white ambulance in Figure 21 with no diversity), as illustrated in Figure 3 and the ImageNet visualizations in the updated appendix, where in all cases Custom Diffusion reconstructs the same instance with limited diversity. In contrast, our method learns a representation that captures the distribution of reference images at set-level in the semantic space. This allows the generated images to be not only consistent with the reference image distribution but also naturally diverse.
> 2. Although manually crafting prompts with a fixed learned embedding can introduce some diversity in generated images, it has the following limitations:
>     - Users must identify specific, describable attributes (e.g., color, background) they wish to modify. In contrast, our method operates in a continuous space, enabling it to model attributes that may not be easily articulated in text.
>     - Users need to experiment with numerous prompts or combinations of attributes (e.g., "red X on a beach," "blue X on a table") to achieve diverse generation results. In contrast, our method automatically models the distribution from the reference images, inferring which attributes will vary without manual intervention.
>     - Again, manual prompt editing does not necessarily ensure that the generated results align with the distribution of the reference image set.
>
>     Moreover, our method retains the ability for text-guided editing, adding an extra layer of flexibility on top of the aforementioned benefits.
>
> 3. Additionally, as shown in Table 1 of our paper, our method shows better classification scores and feature-based scores than other personalization baselines in almost all cases under fair comparison where no additional text promt is used. Due to the limited time for rebuttal, we may not be able to generate a new ImageNet dataset combining diverse text prompts with learned tokens using the personalization baselines, however, we show classification results on a randomly sampled 10-class ImageNet subset, where we generate training images using the Custom Diffusion learned token, randomly combined with diverse CLIP Prompts. The results below demonstrate that our method, without adding text prompts, still outperforms Custom Diffusion with diverse text prompts applied. We will aim to include numbers on full-size dataset in the camera-ready version if the paper is accepted.
>
>
>
> | Method | Top 1 |
> | -------- | -------- |
> | Custom Diffusion + CLIP Prompts    | 89.4 |
> | Ours | 93.1 |
>
>
>
> # Characteristic of Soft Prompts
> > What are the results when only one learned soft prompt is used? Can we confirm what characteristics each soft prompt has learned?
>
> When only one soft prompt is used, our method becomes mathematically equivalent to Textual Inversion. Currently, the optimized soft prompts automatically fit a distribution that represents the reference image set in semantic space, making the characteristics of each prompt not easily interpretable. However, it would be an interesting future direction to explore regularizations or constraints that could control what each prompt learns, offering more flexibility in combining different characteristics for generation.
>
> Notably, while our method can be easily adapted to function equivalently to Textual Inversion by setting the number of prompts to 1, the reverse is not true. Textual Inversion is not suitable for category-level personalization, as shown by the qualitative and quantitative results in Figure 3 and Figure 4. A simple adaptation of Textual Inversion for category-level personalization would involve running it multiple times on different subsets of reference images. However, as demonstrated in Appendix Section C.6, our method still outperforms this adapted approach.
>
>
>
> # Number of Prompt Tokens
> > How the results change according the value of M?
>
> We have added an ablation study on the number of tokens used in each learnable prompt, keeping all other hyperparameters constant. The results are shown in Figure 10 in the updated appendix. Similar to the effect of varying the number of prompts, increasing the number of tokens generally leads to better results, though the effect is less pronounced compared to changing the number of prompts.
>
> Edit: updated figure number

---

> ### Author Response · Authors · 2024-12-02
> **Happy to answer any further questions**
>
> Dear Reviewer RW1y,
>
> Thank you for taking the time to review our submission. We have provided responses to your suggestions and questions and would be happy to address any further concerns or questions. We are looking forward to your feedback.
>
> Best,
>
> Authors of submission 503

---

### Meta-Review · Area_Chair_SAtE · 2024-12-21

**Metareview:**

This paper aims to tackle the problem of the lack of diversity when generating personalized images with pre-trained T2I diffusion models. The proposed method learns a distribution of training texts as a distribution of soft prompts. While the previous works rely on fixed text embedding, the proposed work utilizes a probabilistic approach to enable the generation of novel images by sampling prompts from the learned distribution. Experiments were conducted on text-to-image, text-to-3D, and synthetic dataset generation.

This paper has mixed borderline opinions (three borderline accepts and one borderline reject), while the negative reviewer and one of the positive reviewers did not engage in the discussion despite the reminders from the authors and the AC. Since this paper did not get responses from some of the reviewers, I thoroughly read all the reviews and responses for a fair assessment.

Many reviewers agreed that the target problem and the solution is well described and the experimental results are extensive. On the other hand, the reviewers raised some concerns:

- [3gbS, RW1y, BPjf] The generation quality by DreamBooth may not be good or the comparison may not be fair. E.g., the method should be compared with a more recent method (e.g., IP-Adapter)
- [RW1y] Insufficient participants for human evaluation
- [RW1y, 3gbS , BPjf] Not enough discussion for the learnable prompt space (e.g., Gaussian distribution assumption, the number of learnable prompts)
- [3gbS] The problem setting is not realistic (e.g., many reference images) and learnable token, orthogonal loss, and reparameterization trick are not novel
- [3gbS] Will this method be able to handle new concepts not learned from T2I DM?

The authors addressed most of the concerns raised by the reviewers. Specifically, the authors conducted additional experiments, including generation with fewer samples (5 reference images), ablation study for orthogonal loss, and additional comparisons with IP-Adapter and textual inversion + noise. The authors also increase the number of participants for a more reliable evaluation. After the discussion, the reviewers engaged in the discussion reached a positive consensus.

The concerns raised by Reviewer BPjf look sufficiently addressed by the authors' rebuttal. Specifically, the authors provided a comparison with the IP-Adapter and showed that the proposed method shows better results. Similarly, I think the additional ablation study and analysis look to address the concerns raised sufficiently. The Reviewer RW1y's concerns are also addressed by the authors. The authors clarified the target task and Figure 5, and increased the participant pool for a more reliable assessment.

Overall, the novelty of the proposed method and its empirical contribution look meaningful for addressing challenges in personalized text-to-image generation. Most of the raised concerns by the reviewers were addressed and I don't find any significant issue of this work after reading all the discussions. I recommend accepting this paper.

**Additional Comments On Reviewer Discussion:**

- [3gbS, RW1y, BPjf] The generation quality by DreamBooth may not be good or the comparison may not be fair. E.g., the method should be compared with a more recent method (e.g., IP-Adapter)
- [RW1y] Insufficient participants for human evaluation
- [RW1y, 3gbS , BPjf] Not enough discussion for the learnable prompt space (e.g., Gaussian distribution assumption, the number of learnable prompts)
- [3gbS] The problem setting is not realistic (e.g., many reference images) and learnable token, orthogonal loss, and reparameterization trick are not novel
- [3gbS] Will this method be able to handle new concepts not learned from T2I DM?

The authors addressed most of the concerns raised by the reviewers. Specifically, the authors conducted additional experiments, including generation with fewer samples (5 reference images), ablation study for orthogonal loss, and additional comparisons with IP-Adapter and textual inversion + noise. The authors also increase the number of participants for a more reliable evaluation. After the discussion, the reviewers engaged in the discussion reached a positive consensus.

Reviewer BPjf and RW1y did not engage in the discussion despite the reminders from the authors and the AC. However, as mentioned in the meta-review, the AC thinks most of their concerns were addressed by the authors' responses.

---

### Decision · Program_Chairs · 2025-01-22

Accept (Poster)